# Receptor-mediated cargo hitchhiking on bulk autophagy

Eigo Takeda [1✉], Takahiro Isoda [1,2,3], Sachiko Hosokawa[1], Yu Oikawa[1], Shukun Hotta-Ren[1], Alexander I May [1] & Yoshinori Ohsumi [1✉]

## Abstract

While the molecular mechanism of autophagy is well studied, the cargoes delivered by autophagy remain incompletely characterized. To examine the selectivity of autophagy cargo, we conducted proteomics on isolated yeast autophagic bodies, which are intermediate structures in the autophagy process. We identify a protein, Hab1, that is highly preferentially delivered to vacuoles. The N-terminal 42 amino acid region of Hab1 contains an amphipathic helix and an Atg8-family interacting motif, both of which are necessary and sufficient for the preferential delivery of Hab1 by autophagy. We find that fusion of this region with a cytosolic protein results in preferential delivery of this protein to the vacuole. Furthermore, attachment of this region to an organelle allows for autophagic delivery in a manner independent of canonical autophagy receptor or scaffold proteins. We propose a novel mode of selective autophagy in which a receptor, in this case Hab1, binds directly to forming isolation membranes during bulk autophagy.

**Keywords** Atg8; Autophagy; Hab1; *Saccharomyces cerevisiae*; Selective Autophagy
**Subject Categories** Autophagy & Cell Death; Membranes & Trafficking; Organelles

## Introduction

Autophagy is an intracellular degradation system that is highly conserved among eukaryotes. This process directs cytoplasmic components to the lysosome/vacuole in response to environmental changes, such as starvation (Rabinowitz and White, 2010; Nakatogawa et al, 2009). During autophagy in yeast, an isolation membrane (or phagophore) extends and then closes, resulting in a portion of the cytoplasm being sequestered within a double membrane-bound structure called an autophagosome. The outer membrane of the autophagosome then fuses with the vacuolar membrane, releasing the inner membrane-bound structure (the autophagic body) into the vacuolar lumen (Takeshige et al, 1992). In wild-type (WT) cells,

autophagic bodies are rapidly disrupted, whereas autophagic bodies remain intact inside the vacuole of cells lacking vacuolar proteases or the phospholipase Atg15 (Takeshige et al, 1992; Kagohashi et al, 2023; Harding et al, 1996; Teter et al, 2001).

Two types of autophagy have been described in the literature. In bulk autophagy, cytosolic components are non-selectively delivered to vacuoles, while selective autophagy involves the targeted delivery of organelles or assembled vacuolar enzymes to vacuoles (Takeshige et al, 1992; Klionsky et al, 1992; Hutchins and Klionsky, 2001; Kageyama et al, 2009). Selective autophagy employs autophagy receptor proteins in addition to the core autophagy machinery; these receptor proteins interact with the ubiquitin-like protein Atg8 and the scaffold (or adapter) protein Atg11, which are important for selective autophagy (Suzuki et al, 2007; Zientara-Rytter and Subramani, 2020). On the other hand, bulk autophagy is induced by starvation or treatment with a TORC1 inhibitor and is generally regarded as a non-selective process. Electron microscopy has suggested that the density of ribosomes and glycolytic enzymes in autophagic bodies is comparable to that of the cytosol, indicating random sequestration of cytosolic material (Takeshige et al, 1992; Baba et al, 1994). Although we have previously reported an analysis of cellular materials degraded by autophagy (Suzuki et al, 2014), advances in purification and mass spectrometric techniques now allow a more comprehensive assessment of autophagy cargo.

In this study, we employ our recently developed method for isolating autophagic bodies in budding yeast (Kawamata et al, 2022) to comprehensively analyze the contents of autophagic bodies by mass spectrometry-based proteomics. We uncover a novel protein that is highly enriched in autophagic bodies, suggesting selectivity of cargo sequestration by bulk autophagy. We further explore the mechanism underlying the preferential delivery of cargo to the vacuole by bulk autophagy.

## Results

### Proteomic analyses of autophagic bodies

To understand how proteins are delivered to the vacuole by bulk autophagy, autophagic bodies were analyzed by liquid chromatography-tandem mass spectrometry (LC-MS/MS)

[1]Cell Biology Center, Institute of Innovative Research, Tokyo Institute of Technology, Yokohama, Japan. [2]School and Graduate School of Bioscience and Biotechnology, Tokyo Institute of Technology, Yokohama, Japan. [3]Frontier Research Center, POLA Chemical Industries Inc., Yokohama, Japan. ✉E-mail: takeda@igm.hokudai.ac.jp; ohsumi.y.aa@m.titech.ac.jp

(Fig. 1A). To this end, we employed our recently established method to obtain intact autophagic bodies from rapamycin-treated or nitrogen-starved *atg15∆* cells (Kawamata et al, 2022). For proteomic analyses, total cell lysates, autophagic body fractions, and control fractions were subjected to label-free quantitative mass spectrometry (Fig. 1B).

Known selective cargoes (Ape1, Ams1, and Lap3), autophagy receptors (Atg19 and Atg34), and autophagy machinery proteins (Atg8 and Atg1) were enriched in autophagic body fractions (Fig. 1C,D, red) (Klionsky et al, 1992; Scott et al, 2001; Kirisako et al, 1999; Kageyama et al, 2009; Nakatogawa et al, 2012; Suzuki et al, 2010; Yuga et al, 2011). On the other hand, histones (Hta1/2, Htb1/2, Hhf1/2, Hht1/2, Htz1, and Hho1), which localize to the nucleus and many glycogen-related enzymes (Gsy1, Gsy2, Glc3, Gph1, and Gbp1), which we recently found are non-preferred cargoes of autophagy (Isoda et al, 2024, manuscript submitted), were detected at very low levels in autophagic bodies (Fig. 1C,D, purple and blue, respectively). These results validate our strategy to evaluate autophagy cargoes.

During these analyses, we identified a yet uncharacterized protein, Ybr285w, that is highly enriched in autophagic body fractions of both rapamycin-treated and nitrogen-starved cells (Fig. 1C,D, black bold). Ybr285w is a small protein comprising 144 amino acid residues without any identifiable membrane-spanning regions. Due to its high degree of enrichment, we named this protein Hab1 (Highly enriched in Autophagic Bodies) and set out to investigate the mechanism of Hab1 enrichment, as well as its function in autophagy.

## Hab1 is preferentially delivered to the vacuole via autophagy

To confirm that Hab1 is preferentially delivered to the vacuole by autophagy, we first employed the GFP cleavage assay to compare the delivery of Hab1 to that of Pgk1, a non-selective cargo (Welter et al, 2010; Iwama and Ohsumi, 2019) and the preferentially degraded proteins Fas1 and Ald6 (Shpilka et al, 2015; Onodera and Ohsumi, 2004). When expressed in fusion with GFP, delivery of a protein to the vacuole yields a vacuolar protease-resistant 'free' GFP moiety (GFP′) (Shintani and Klionsky, 2004). Free GFP, which can be detected by immunoblotting, was compared to the amount of the full-length (non-delivered) protein. More than 50% of Hab1 was delivered to the vacuole after 4 h of rapamycin treatment or nitrogen starvation. This shows that Hab1 delivery was markedly higher than that observed for other proteins (<10% each; Fig. 2A), which is consistent with our proteomic data (Fig. 1C,D). Moreover, the expression of Hab1 was induced by both rapamycin treatment and nitrogen starvation, which is in line with reports of comprehensive transcriptome analyses (Hughes Hallett et al, 2014; Gasch et al, 2000; Makino et al, 2021).

Next, we observed Hab1-GFP by fluorescence microscopy. In growing cells, a faint Hab1-GFP signal was observed in the cytosol (Fig. 2B). After 1 h of rapamycin treatment, the cytosolic Hab1-GFP signal increased and Hab1-GFP puncta appeared adjacent to the vacuole in WT cells (Figs. 2B and EV1A). Few puncta were observed in *atg2∆* or *atg1∆* cells. Hab1-GFP puncta colocalized with mCherry-Atg8, a marker of the pre-autophagosomal structure (PAS) (Fig. EV1B) (Suzuki et al, 2007). Time-lapse imaging showed that Hab1 assembles to the PAS later than Atg8, indicating that

Hab1 is recruited after PAS formation (Fig. EV1C). At 4 h rapamycin treatment, the GFP signal was predominantly observed within the vacuole of WT cells, while it was retained in the cytosol in *atg2∆* and *atg1∆* cells (Figs. 2B and EV1A).

Next, we assessed the dependence of Hab1-GFP cleavage on Atg proteins during rapamycin treatment or nitrogen starvation. Free GFP appeared under both autophagy-inducing conditions in WT cells, but was not detected in *atg1∆*, *atg2∆*, or *atg15∆* cells (Fig. 2C; Appendix Fig. S1). These results indicate that Hab1 is delivered to the vacuole by autophagy. We further confirmed that core Atg proteins (Atg1, Atg2, Atg3, Atg8, Atg9, Atg13, Atg14, and Atg17) were required for the delivery of Hab1 to the vacuole (Fig. 2D), as was Atg24, which is required for autophagy of organelles and large complexes such as ribosomes (Shatz et al, 2024; Kanki and Klionsky, 2008; Nemec et al, 2017; Shpilka et al, 2015). On the other hand, Atg11, the scaffold protein implicated in selective autophagy, was not (Fig. 2D).

## The N-terminal region of Hab1 is necessary for preferential delivery

We next sought to characterize the Hab1 protein and identify relevant sub-domains by creating a series of Hab1 truncation mutants. First, C-terminal truncations of Hab1 were constructed (Fig. 3A). To our surprise, the delivery of incremental truncation mutants down to and including Hab1(1–42) was almost the same as that of the full-length protein, whereas that of Hab1(1–38) was severely impaired (Fig. 3B). This indicates that a very small subset of Hab1 is sufficient for preferential degradation via autophagy. Meanwhile, N-terminal truncation mutants of Hab1(19–144, 31–144, 43–144) were only marginally delivered to the vacuole, suggesting that the N-terminal residues are also essential for preferential delivery (Fig. EV2A). In this region, we identified an amphipathic helix (3–15) (Fig. 3C), as predicted by HeliQuest (Gautier et al, 2008). In some cases, amphipathic helices interact with membranes via hydrophobic residues predominant on one side of the helix (Drin and Antonny, 2010; Kotani et al, 2018; Mochida et al, 2022), suggesting that the N-terminal helix of Hab1 might bind to a membrane. Indeed, we observed that Hab1(1–18)-GFP localizes to the endoplasmic reticulum and other membranes (Figs. 3D and EV2B), while a mutant for the hydrophobic side of the amphipathic helix (1–18, 3A) lost this membrane association (Fig. 3D). Furthermore, Hab1(3A)-GFP was defective in its delivery to the vacuole (Fig. 3D,E), showing that membrane binding of the N-terminal amphipathic helix is important for the preferential delivery of Hab1.

Various autophagy receptor proteins contain an Atg8-family interacting motif (AIM) (Pankiv et al, 2007; Noda et al, 2008, 2010). We identified a candidate AIM (Phe[35]–Ile[36]–Asn[37]–Ile[38]) in Hab1(1–42) (Fig. 3F). To determine whether this sequence functions as a bona fide AIM, the delivery of Hab1 variants harboring mutations within the putative AIM (F35A, I38A, and a F35A/I38A double mutant) was examined. In all three AIM mutants, GFP cleavage upon rapamycin treatment was notably decreased in comparison with WT Hab1 (Fig. 3G). Further, we confirmed that the AIM mutation impaired both Hab1(1–42) assembly and delivery to vacuoles (Fig. EV2C), indicating that the AIM plays an essential role in the preferential delivery of Hab1.

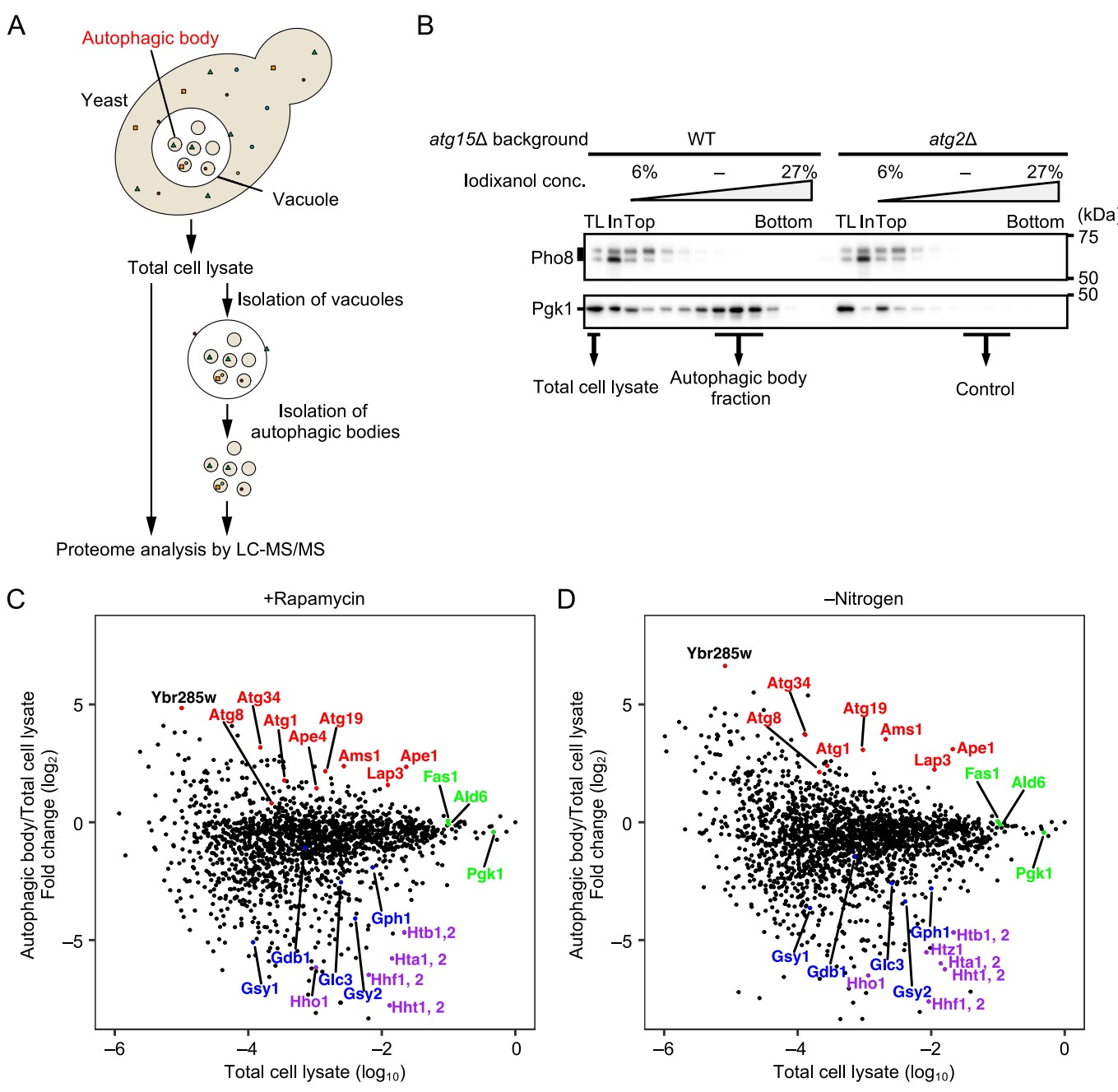

**Figure 1. Proteomic analyses of autophagic bodies.**

(A) Overview of the strategy used to isolate autophagic bodies and analyze their cargo. (B) Immunoblots of fractionated autophagic body samples. Logarithmically growing *atg15Δ* cells in SD medium were treated with rapamycin (200 ng/mL) for 2 h. Autophagic bodies were isolated from disrupted vacuoles by density gradient centrifugation and subjected to immunoblotting. Heavy fractions containing Pgk1 indicate autophagic body fractions. Control fractions were obtained from *atg15Δ atg2Δ* cells by the same procedure. Pho8 was used as a vacuolar marker. TL, total cell lysate; In, input. (C, D) The logarithm of protein enrichment in isolated autophagic bodies (C) 2 h rapamycin treatment; (D) 2 h nitrogen starvation) (y-axis) and the logarithm of abundance in total cell lysates (x-axis) are indicated. Red, known preferential cargo and autophagy machinery proteins; green, cytosolic proteins described in the main text; blue, glycogen-related enzymes; purple, histones. Source data are available online for this figure.

## Hab1(1–42) specifically binds to lipidated Atg8

We next investigated the interaction between Hab1(1–42) and Atg8. Atg8 coprecipitated with Hab1(1–42) but not with an AIM mutant (F35A). Remarkably, we found that Atg8 binding to

Hab1(1–42) occurred exclusively with the lipidated form, Atg8-PE (Fig. 4A). Moreover, the disruption of *ATG4*, *ATG3*, or *ATG5*, which are necessary for the lipidation of Atg8 (Ichimura et al, 2000; Kirisako et al, 2000; Hanada et al, 2007), abolished the interaction between Atg8 and Hab1 (Fig. 4B), indicating that Hab1(1–42)

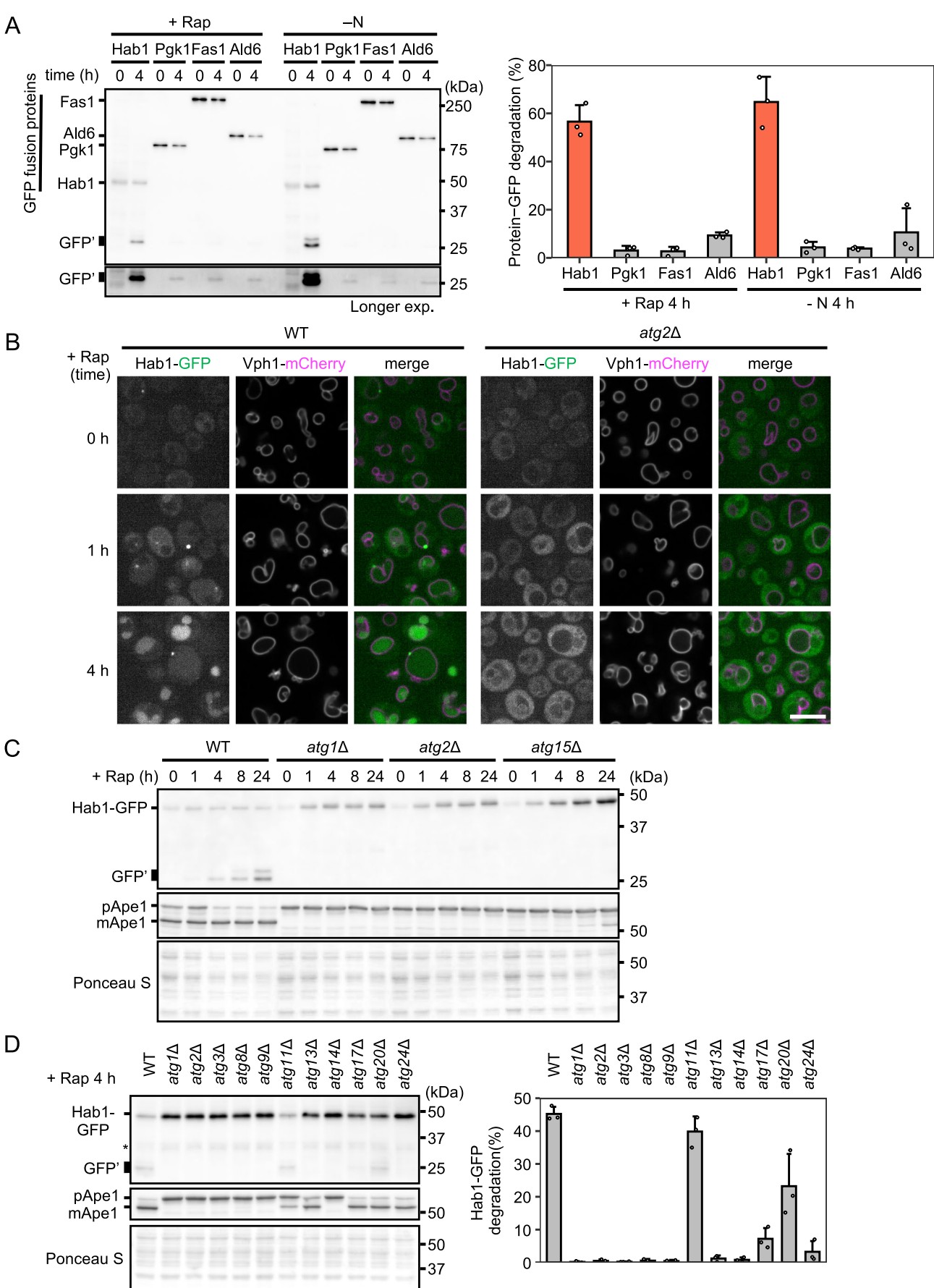

◄

**Figure 2. Hab1 is preferentially delivered to the vacuole via autophagy.**

(A) GFP cleavage of Hab1-GFP, Pgk1-GFP, Fas1-GFP, and Ald6-GFP. Each sample was examined after appropriate dilution (Pgk1-GFP: 1/500, Fas1-GFP: 1/50, Ald6-GFP: 1/500). GFP', free-GFP; +Rap, rapamycin treatment; –N, nitrogen starvation. Intensities of protein-GFP and GFP' bands were measured to estimate the delivery of protein-GFP to vacuoles. The percentages of GFP' to total GFP (GFP' + full-length GFP) are shown as mean ± SD (n = 3). (B) Confocal fluorescence imaging of WT or atg2Δ cells expressing Hab1-GFP and Vph1-2xmCherry (a vacuolar membrane protein). Scale bar, 5 μm (C) GFP cleavage of Hab1-GFP was determined in WT, atg1Δ, atg2Δ, and atg15Δ cells at indicated time points of rapamycin treatment. Ape1 is shown as a marker of autophagic activity. Total protein was visualized by Ponceau S staining. (D) Dependence of Hab1 degradation on Atg proteins, as determined by Hab1-GFP cleavage following rapamycin treatment. * = intermediate degradation or mid-translation products of Hab1-GFP. Percentages of GFP' to total GFP (GFP' + full-length GFP) are shown as mean ± SD (n = 3). Blot data are representative of two (C) or three (A, D) independent experiments. Source data are available online for this figure.

specifically binds to Atg8-PE. We also confirmed that full-length Hab1 binds specifically to the lipidated form of Atg8 (Appendix Fig. S2).

Atg8-PE is known to be the only protein that tightly binds to autophagic membranes such as the isolation membrane and autophagosomes (Ichimura et al, 2000; Kirisako et al, 2000). However, no receptor that binds specifically to Atg8-PE has been reported to date. In order to confirm that the N-terminal amphipathic helix of Hab1 is responsible for the specific recognition of Atg8-PE, we examined interactions between Atg8-PE and Hab1(19–42), an N-terminal helix truncate, and Hab1(3A), a mutant for the hydrophobic side of the helix. Both mutant forms completely abolished the interaction with Atg8-PE, although weak interaction with the non-lipidated form of Atg8 was observed (Fig. 4C). This indicates that the N-terminal helix of Hab1 is essential for its specific interaction with Atg8-PE. Taken together, these results indicate that the AIM and amphipathic helix within a small region of Hab1 cooperatively function to bind Atg8-PE, thereby facilitating delivery by autophagy.

## A possible role for Hab1 in ribosomal degradation

Truncation mutants of Hab1 showed no significant change in Ape1 processing, and autophagy activity was not markedly impaired in Hab1-deficient cells (Fig. 3B,E,G; Appendix Fig. S3), indicating that Hab1 is likely not involved in the tuning of autophagy activity per se. Consequently, we considered that Hab1 might act as an autophagy receptor for an unidentified cargo. We found that the disruption of Atg24 decreased Hab1 delivery (Fig. 2D); this protein has recently been reported to be necessary for autophagic degradation of large cargoes by ensuring that a large opening pore in expanding isolation membranes is sufficiently open to accommodate bulky material (Kotani et al, 2023). This result implies that Hab1 binds an unidentified large cargo. In line with this, we found that disruption of Atg24 markedly decreased the degradation of Hab1(43–144) but not Hab1(1–42) upon rapamycin treatment (Fig. 5A), suggesting that the C-terminal region of Hab1 is the cargo binding region.

To determine the interaction partner, we performed immunoprecipitation of GFP-Hab1(43–144) and analyzed associated proteins by LC-MS/MS. A total of 66 proteins were identified, among which ribosomal proteins showed high enrichment (23 proteins; Figs. EV3A and EV3B). This suggests that Hab1 binds ribosomes via its C-terminal region. To further confirm this finding, we subjected a cell lysate from Hab1 truncation mutants to ultracentrifugation at $100,000 \times g$ to sediment ribosomes. Full-length Hab1 and Hab1(43–144) were detected in the pellet, while Hab1(1–42) was recovered in the supernatant (Fig. 5B), providing further evidence that the C-terminal region of Hab1 binds to ribosomes.

We also fractionated lysates of rapamycin-treated cells expressing Hab1-GFP by sucrose density gradient centrifugation after treatment with a cross-linking reagent. Immunoblotting of these samples revealed the presence of Hab1-GFP, Rpl23 (60S ribosomal protein), and Rps25B-HA (40S ribosomal protein), and spectrophotometry of fractionated samples showed that Hab1-GFP was detected in fractions correlating with 80S ribosome and 60S ribosomal subunit absorbance peaks (Fig. 5C), but not in the 40S ribosomal subunit fractions. Together, these results indicate that Hab1 binds to ribosomes via the 60S ribosomal subunit.

To determine whether Hab1 acts as an autophagy receptor for ribosomes, we examined how deletion or overexpression of Hab1 affects the autophagic delivery of ribosomes to the vacuole. Autophagic bodies obtained from atg15Δ, hab1Δ atg15Δ cells, or atg15Δ cells overexpressing Hab1 under the control of the TEF1 promoter (Figs. EV3C and EV3D) (Janke et al, 2004) following 6 h rapamycin treatment were analyzed by LC-MS/MS (Fig. 5D). Most ribosomal proteins in autophagic bodies isolated from hab1Δ cells showed a decrease in abundance (approximately 5%) compared with WT cells (Fig. 5D, left). On the other hand, the majority of ribosomal proteins in Hab1-overexpressing cells were enriched in autophagic bodies when compared with WT cells, with about 25% and 15% increases in the amount of 60S- and 40S-subunit proteins observed, respectively (Fig. 5D, right). These results show a correlation between the amount of Hab1 and ribosome delivery to the vacuole. Among the non-ribosomal proteins identified by immunoprecipitation (Fig. EV3A), no other protein showed robust delivery to the vacuole in a Hab1-dependent manner (Fig. EV3E), suggesting that ribosome is the major cargo of Hab1.

Further, the relative increase in delivery of 60S subunit proteins vis-à-vis 40S subunit proteins provides another indication that the 60S subunit contains a binding site for Hab1, resulting in free 60S subunit delivery to the vacuole in Hab1-overexpressing cells. We found that even after 4 h nitrogen starvation, a condition under which Hab1 is highly expressed, the molecular abundance of Hab1 was 1.25% to 2.5% of that of ribosomes (Fig. EV3F). Due to the greater number of ribosomes in comparison to Hab1, these results suggest that ribosomes are delivered mostly by non-selective autophagy rather than the Hab1-mediated process.

We also performed transmission electron microscopy (TEM) to study the localization of ribosomes in autophagic bodies after 2 h rapamycin treatment. Ribosomes, which are visualized as dots of high electron density, were enriched on the inner surface of autophagic body membranes in Hab1-overexpressing cells (Fig. 5E, arrowheads). Ribosomes associated with autophagic body membranes were quantified, revealing an approximately 25% reduction in autophagic body membrane-association in hab1Δ cells in comparison to WT. In contrast, Hab1-overexpression resulted in a ~60% increase (Fig. 5F). The Hab1-dependent binding of ribosomes to the membrane of

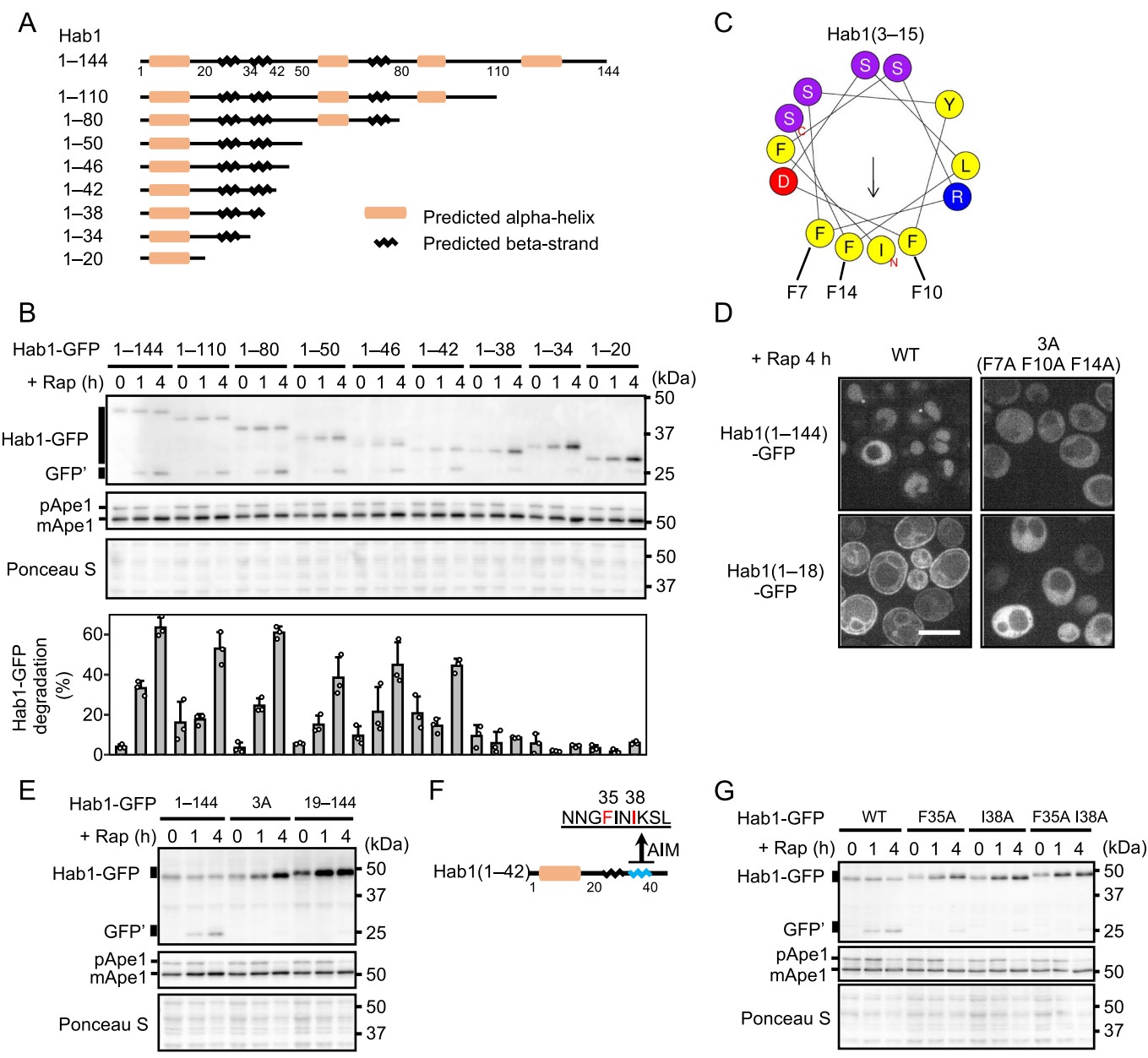

**Figure 3. The N-terminal amphipathic helix and the AIM of Hab1 are responsible for preferential delivery.**

(A) Secondary structure of Hab1 as predicted by JPred 4. Truncation mutants of Hab1 are also depicted. (B) GFP cleavage of Hab1 C-terminal truncation variants. Percentages of GFP' to total GFP (GFP' + full-length GFP) are shown as mean ± SD (n = 3). (C) A helical wheel projection of residues 3-15 of Hab1. (D) Confocal fluorescence imaging of GFP-tagged Hab1 variant-expressing cells after 4 h rapamycin treatment. Scale bar, 5 μm. (E) GFP cleavage assay of Hab1 N-terminal helix mutants. (F) An illustration of the secondary structure of Hab1(1–42). (G) GFP cleavage of WT and putative AIM mutants (F35A, I38A, and F35A/I38A) of Hab1. Blot data are representative of two (E, G) or three (B) independent experiments. Source data are available online for this figure.

autophagic bodies is consistent with the notion that Hab1 binds to autophagic membranes via Atg8-PE and indicates that Hab1 brings ribosomes to autophagic membranes.

## The N- and C-terminal regions of Hab1 are important for the delivery of ribosomes to vacuoles

To determine the region responsible for Hab1 binding to ribosomes, we focused on sequence conservation in the

C-terminal region. Using PSI-BLAST searches, we identified putative Hab1 orthologs among *Saccharomycetaceae* species (Fig. EV4A), including *Zygosaccharomyces rouxii* (GAV56356.1), *Torulaspora delbrueckii* (XP_003680961.1), *Lanchancea thermotolerans* (XP_002556312.1), and *Kluyveromyces marxianus* (XP_022675130.1). Since we identified amino acid conservation of several amino acid residues within regions 50–70 and 110–140 of Hab1 (Fig. EV4A, red), we constructed deletion mutants lacking these regions: Hab1(70–144) and Hab1(43–110). The binding of

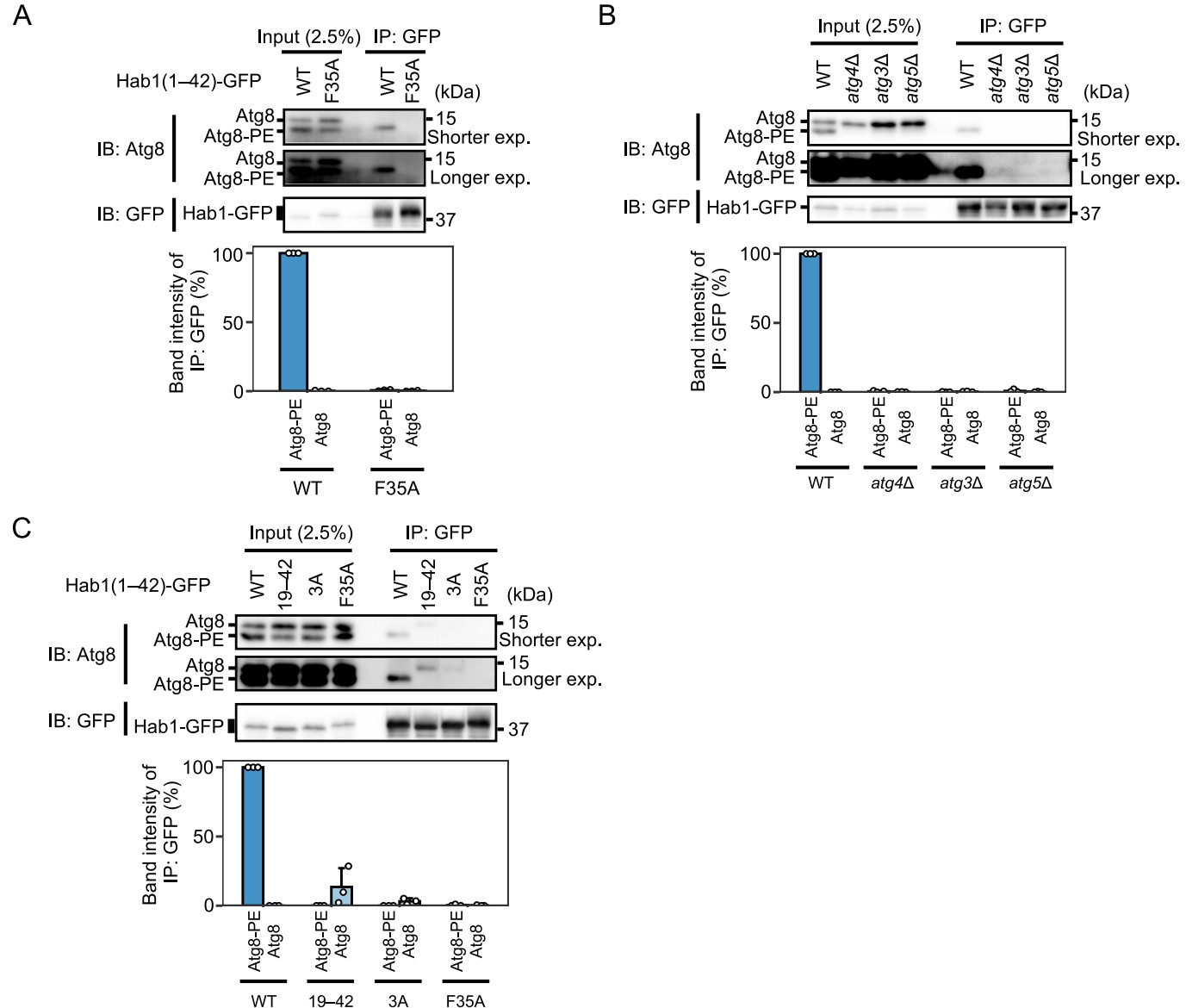

**Figure 4. Hab1(1–42) specifically binds to lipidated Atg8.**

(A) Binding of Atg8 to Hab1(1–42)-GFP as determined by immunoprecipitation. Samples were obtained from cells treated with rapamycin for 1 h before being subjected to immunoblotting. Quantifications of the immunoprecipitation band intensities are shown as mean ± SD ($n = 3$); data are normalized to the WT Atg8-PE band. (B) Binding of Atg8 to Hab1(1–42)-GFP in Atg8 lipidation-defective mutants assessed by immunoprecipitation. Quantifications of band intensities are shown as mean ± SD ($n = 3$); data are normalized to the WT Atg8-PE band. (C) Binding of Atg8 to Hab1(1–42)-GFP variants assessed by immunoprecipitation. Quantifications of band intensities are shown as mean ± SD ($n = 3$); data are normalized to the WT Atg8-PE band. Blot data are representative of three independent experiments (A–C). Source data are available online for this figure.

these mutants to ribosomes was assayed by ultracentrifugation ($100,000 \times g$), which revealed that ribosome binding was partially decreased in Hab1(43–110) and almost completely absent in Hab1(70–144) (Fig. EV4B). These results indicate that the 43–70 region of Hab1 is required for binding to the ribosome.

We next introduced mutations at conserved residues (D52, E57, M58) of Hab1. Subcellular fractionation revealed that the M58A mutation significantly decreased the binding of Hab1 to the ribosome (Fig. EV4C). Cleavage assay analyses of these mutants indicates that the M58A mutant, which does not bind ribosomes, was delivered to the vacuole in the absence of Atg24 (Fig. EV4D).

This suggests that binding of Hab1 to ribosomes, which are relatively large, is responsible for the impaired Hab1 delivery observed in *atg24Δ* cells. To confirm the contribution of the N- and C-terminal regions of Hab1 to cargo delivery, we performed cargo analyses and found that the expression of Hab1(WT) in *hab1Δ* cells increases ribosome delivery to vacuoles. As this effect was not observed upon expression of Hab1(43–144) or Hab1(M58A) (Fig. EV4E), these results show that both the N-terminal region and C-terminal ribosome binding region of Hab1 are important for Hab1-dependent delivery of ribosomes to the vacuole.

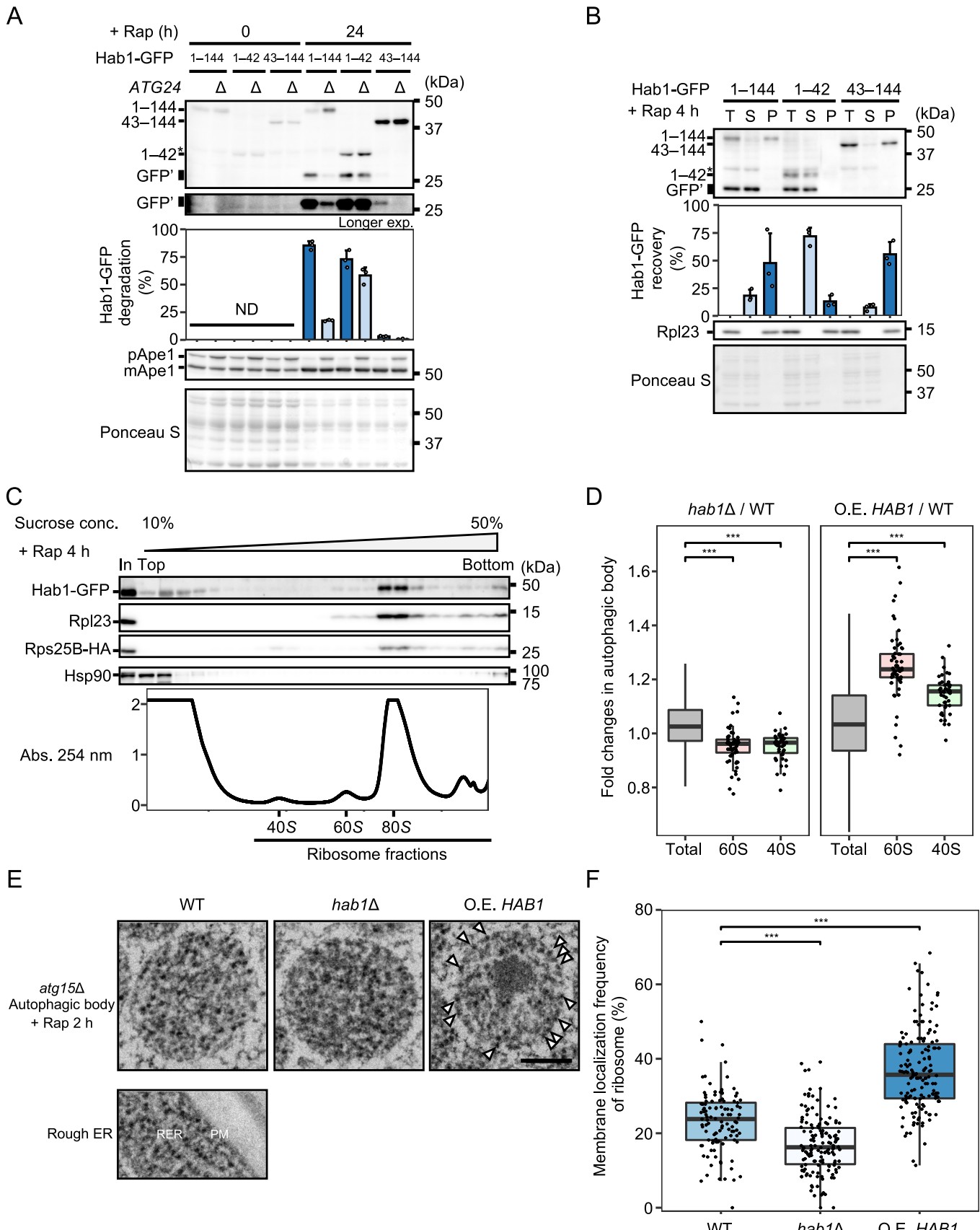

◄

**Figure 5. A possible role for Hab1 in ribosomal degradation.**

(A) GFP cleavage of Hab1 truncation mutants in WT or atg24Δ cells. Percentages of GFP' to total GFP (GFP' + full-length GFP) are shown as mean ± SD ($n = 3$). ND, not determined. (B) Immunoblotting of subcellular fractionated lysates of Hab1-GFP strains. Total (T), supernatant (S), and pellet (P) fractions from cell lysates after rapamycin treatment of cells are shown. Rpl23 is a ribosomal protein control. Percentages of Hab1-GFP recovery of S and P (relative to T) are shown as mean ± SD ($n = 3$). (C) Immunoblots of sucrose density gradient centrifugated fractions after cross-linking using DSP. Lysates were obtained from atg2Δ cells expressing Hab1-GFP following 4 h of rapamycin treatment. Rpl23 and Rps25B-HA were used as markers of the 60S and 40S ribosomal subunits, respectively. Hsp90 is a negative control that does not bind to ribosomes. Absorbance at 254 nm reflects the amount of RNA. In, input. (D) Proteomic analyses of ribosomal proteins in autophagic bodies. Autophagic bodies isolated from atg15Δ, hab1Δ atg15Δ, and HAB1-overexpressing atg15Δ cells following 6 h rapamycin treatment were subjected to proteomic analyses. Fold changes are shown as normalized label-free quantification of each 40S or 60S ribosomal protein in autophagic bodies isolated from hab1Δ or HAB1-overexpressing cells relative to that of WT cells. "Total" indicates all detected proteins. ***$p < 0.001$. $p$ values were calculated using an unpaired two-sided Welch's t-test. Box plots are shown as median (middle bar) with 25th and 75th percentiles and 1.5 × interquartile ranges in whiskers. The numbers of annotated proteins are as follows: Total, 1888; 60S, 59; 40S, 47. (E) Representative TEM images of autophagic bodies in atg15Δ, atg15Δ hab1Δ, and HAB1-overexpressing atg15Δ cells. A TEM image of the rough endoplasmic reticulum (RER) is shown as a morphological control for ribosomes. PM, Plasma membrane. Scale bar, 200 nm. (F) Quantification of the frequency of ribosomes associated with autophagic body membranes, related to (E). The membrane localization frequencies of ribosomes in individual autophagic bodies are shown as dots in box plots. The $n$ values for each sample are as follows: WT, 109; hab1Δ, 146; HAB1-overexpression, 151. ***$p < 0.001$. $p$ values were calculated using an unpaired two-sided Welch's t-test. Box plots are shown as median (middle bar) with 25th and 75th percentiles and 1.5 × interquartile ranges in whiskers. Blot data are representative of two (C, D) or three (A, B) independent experiments. Source data are available online for this figure.

## Hab1(1–42) functions as an autophagy degron

In general, autophagy receptors are thought to recognize and bind cargo, ultimately delivering both the receptor and cargo to the vacuole for degradation. However, the preferential degradation of Hab1 was only slightly affected in mutants lacking the C-terminal region that binds to the ribosome (Fig. 5B). This suggests that the N-terminal region of Hab1 itself may function as an autophagy degron. We tested this possibility by synthetically binding Hab1(1–42) to Pho8Δ60, a cytosolic proenzyme that is processed to an active form by vacuolar proteases upon delivery to the vacuole (Noda et al, 1995) thereby providing a quantitative measure of delivery through the evaluation of Pho8 phosphatase activity. We engineered a synthetic interaction between Pho8Δ60 and Hab1(1–42) by expressing Hab1(1–42)-GBP and Pho8Δ60-GFP fusion proteins (Fig. 6A); the GBP (GFP-binding protein) nanobody strongly binds GFP in vivo (Rothbauer et al, 2006; Ukai et al, 2018). These cells were treated with rapamycin for 4 h and the delivery of Pho8Δ60-GFP was assessed by the ALP assay. The ALP activity of Hab1(1–42)-GBP expressing cells was markedly higher than that of Hab1(1–42)-GBP3A (a variant of GBP that does not bind GFP) (Fig. 6B). Moreover, ALP activity was low in atg2Δ cells expressing Hab1(1–42)-GBP (Fig. 6B), demonstrating that Hab1(1–42)-bound Pho8Δ60 is preferentially delivered to the vacuole in an autophagy-dependent manner. Fluorescence microscopy also confirmed the highly efficient delivery of Hab1(1–42)-GBP-bound Pho8Δ60-GFP (Fig. 6C). Together, these data indicate that the association of Hab1(1–42) with a cytosolic protein is sufficient to cause the preferential targeting of the latter by autophagy.

Finally, we examined whether Hab1(1–42) is able to mediate the delivery of a much larger cargo, such as an organelle, to the vacuole. To this end, we employed the ALFA-tag system (Götzke et al, 2019) to bind Hab1(1–42) to mitochondria by tagging Tom70 with an ALFA-tag that specifically associates with Hab1(1–42)-NbALFA (anti-ALFA-tag nanobody). Om45-GFP cleavage was used to monitor delivery of mitochondria to the vacuole in these experiments (Fig. 6D). After 24 h autophagy induction by nitrogen starvation, free GFP, which reflects the delivery of mitochondria to the vacuole, was detected in WT cells. Delivery of this organelle was increased by expression of Hab1(1–42)-NbALFA when compared with WT cells expressing Hab1(1–42) without NbALFA

(Figs. 6E and EV5A). Furthermore, the expression of Hab1(1–42)-NbALFA recovered mitochondrial delivery in the absence of the mitophagy receptor Atg32 (Figs. 6E and EV5A) (Kanki et al, 2009; Okamoto et al, 2009).

Autophagic degradation of most organelles, such as mitochondria, requires the scaffold protein Atg11. To examine whether Hab1(1–42) mediated organelle degradation is dependent on Atg11, we expressed Hab1(1–42)-NbALFA in atg11Δ and atg32Δ atg11Δ strains. The expression of Hab1(1–42)-NbALFA clearly recovered delivery of mitochondria in both atg11Δ and atg32Δ atg11Δ strains (Figs. 6F and EV5A), indicating that Hab1 selectively delivers cargo via an Atg11-independent mechanism. Similar results were obtained when assessing Hab1-mediated delivery of peroxisomes, demonstrating the generality of this phenomenon (Figs. EV5B, EV5C, and EV5D). In conclusion, Hab1(1–42) functions as an autophagic degron allowing the degradation of organelles in an Atg11-independent manner, illustrating that Hab1 can bypass the canonical receptor/scaffold-mediated pathway of selective autophagy.

## Discussion

In this study, we examined the proteome of autophagic cargo using cell fractionation and mass spectrometry. Uniquely, our proteomic approach allowed the direct and comprehensive investigation of protein delivery to the vacuole by autophagy. There was no striking discrepancy in cargo selectivity between rapamycin- and nitrogen starvation-induced forms of autophagy. Rather, we found that the amount of a protein delivered by autophagy is correlated with its abundance in total cell lysates both during rapamycin treatment and nitrogen starvation, although there is some variability (Fig. 1C,D). We interpret these data as reflecting the generally random nature of sequestration via bulk autophagy. However, we also uncovered an uncharacterized protein that is preferentially degraded by autophagy. This protein, Hab1, is likely an Atg11-independent receptor that is involved in the selective removal of a subset of ribosomes.

Most receptor proteins are known to bind both Atg8 and Atg11 in yeast (Farré et al, 2008; Okamoto et al, 2009; Kanki et al, 2009; Mochida et al, 2015; Wilfling et al, 2020; Shintani et al, 2002; Zientara-Rytter and Subramani, 2020; Motley et al, 2012; Suzuki

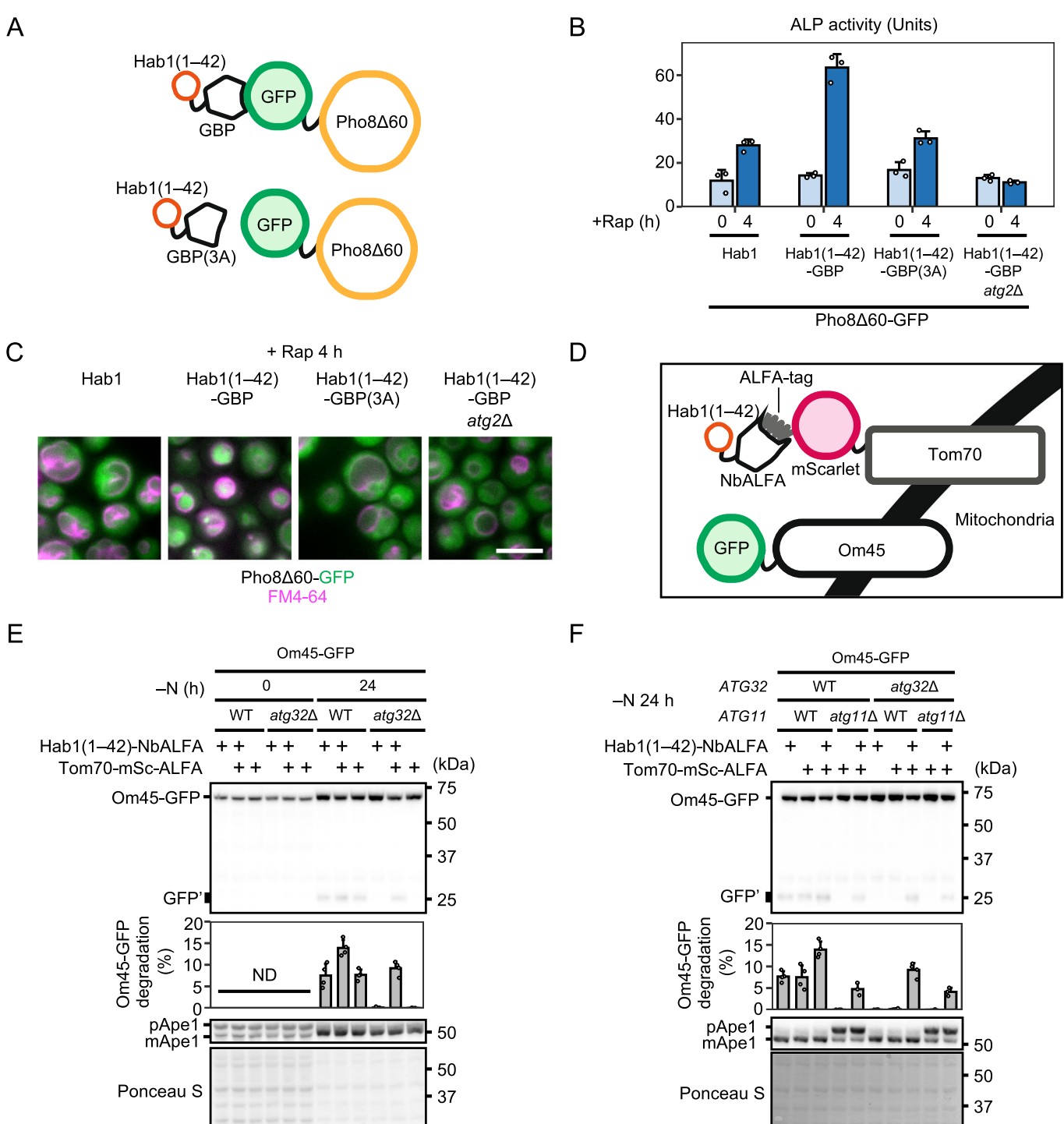

**Figure 6.  Hab1(1–42) functions as an autophagy degron.**

(A) An overview of the strategy used to determine Pho8Δ60-GFP delivery by binding to Hab1(1–42)-GBP. (B) The effect of Hab1(1–42) binding to Pho8Δ60 on its delivery as determined by ALP assay. Data were acquired from biological triplicates (mean ± SD). (C) Fluorescence micrographs of Pho8Δ60-GFP-expressing cells. Merged images shown. FM4-64, vacuolar marker. Scale bar, 5 μm. (D) An illustration of the organelle delivery assay employing synthetic tethering using the ALFA-tag system. (E) Rates of mitochondrial delivery to the vacuole were examined by GFP cleavage of Om45-GFP. Hab1(1–42) was tethered to mitochondria using the ALFA-tag system. –N, nitrogen starvation; mSc, mScarlet (Bindels et al, 2017); Percentages of GFP' to total GFP (GFP' + full-length GFP) are shown as mean ± SD (n = 3). ND, not determined. Blot data are representative of two independent experiments. (F) Rates of mitochondrial delivery to the vacuole were examined by GFP cleavage of Om45-GFP. Percentages of GFP' to total GFP (GFP' + full-length GFP) are shown as mean ± SD (n = 3). A subset of data is derived from samples used for quantification in (E). Blot data are representative of three independent experiments. Source data are available online for this figure.

et al, 2010). The molecular mechanism of Atg11 function in selective autophagy has been well studied, with Atg11 thought to function as a scaffold protein for the assembly of Atg proteins during selective autophagy (Zientara-Rytter and Subramani, 2020). The interactions between receptor proteins and Atg11 or Atg8 are regulated by post-translational modifications of receptor proteins, such as phosphorylation (Kanki et al, 2013; Farré et al, 2013; Pfaffenwimmer et al, 2014; Tanaka et al, 2014). The binding of a receptor protein to Atg11 activates Atg1 kinase, which assembles Atg proteins to form the PAS (Kamber et al, 2015; Torggler et al, 2016), thereby allowing the selective sequestration of the receptor and its cargo into autophagosomes (cargo-induced selective autophagy) (Turco et al, 2020). Recently, numerous mammalian autophagy receptors (p62, CCPG1, NDP52, NBR1, Optineurin, and TAX1BP1) were found to interact with FIP200 (an ortholog of Atg11) via their FIP200-interacting region (Vargas et al, 2019; Ravenhill et al, 2019; Smith et al, 2018; Turco et al, 2019; Ohnstad et al, 2020; Turco et al, 2021; Zhou et al, 2021), indicating that cargo-induced selective autophagy is conserved from yeast to mammals.

While Hab1 was highly enriched in autophagic bodies in the absence of Atg11 (Fig. 2D), many other enriched proteins are known to be delivered to the vacuole in an Atg11-dependent manner (Fig. 1C,D; Ape1, Ams1, Lap3, Atg19, and Atg34). Unlike cargo-induced selective autophagy, we find that Hab1 is preferentially sequestered into autophagosomes by its binding of Atg8-PE on isolation membranes (Fig. 4A) (Kirisako et al, 1999). This suggests that in contrast to conventional receptor proteins, Hab1 is not involved in the initiation of PAS formation; rather, it specifically binds to autophagic membranes that are already forming to afford a degree of selectivity to bulk autophagy. This unique function of Hab1 depends on its short N-terminal region. It is possible that the amphipathic helix of Hab1 recognizes lipid membranes and the AIM of Hab1 recognizes Atg8, resulting in strong binding exclusively to Atg8-PE. Our finding that Hab1 is recruited to the PAS in a core Atg-dependent manner is consistent with a model whereby Hab1 functions after the initiation of bulk autophagy (Figs. 2B and EV1C).

Given these findings, we propose a novel form of selective autophagy we refer to as "cargo hitchhiking". This type of autophagy is initiated on isolation membranes formed during the autophagy process. The synthetic association of the autophagy degron region of Hab1 (residues 1–42) with a range of cellular components is sufficient to allow efficient degradation via this pathway. There may exist receptors that specifically interact with Atg8-PE in mammalian cells as well: artificial autophagosome markers have been reported, consisting of combinations of an AIM and a hydrophobic sequence or a lipid-binding region (Lee et al, 2017; Stolz et al, 2017). This indicates that receptor protein-isolation membrane interactions may manifest in a range of eukaryotes.

Our results also shed light on the degradation of ribosomes by autophagy. Previous studies have reported Ubp3/Bre5-dependent selective degradation of the 60S ribosomal subunit by autophagy, but the detailed mechanism behind this process, including whether it is regulated by receptor proteins, remains unclear (Kraft et al, 2008). We and other groups have shown that disruption of UBP3 or BRE5 causes a partial defect in bulk autophagy (Huang et al, 2015; Müller et al, 2015). Whether autophagic degradation of ribosomes is a genuinely selective process remains to be determined.

In terms of the amount of ribosomal proteins degraded, Hab1-mediated ribosomal degradation appears to be limited when compared with Hab1-independent degradation. As ribosomes are abundant and bulk autophagy is highly active, the amount of ribosome non-selectively degraded by bulk autophagy may be much greater than those by Hab1-mediated process (Figs. 5D and EV3F). It is possible that Hab1 regulates specific ribosomal functions through the selective removal of a subset of ribosomes. Ribosomes are highly diverse: translation state, differences in translated mRNAs, ribosomal modifications, subunit dropouts, and subunit variations all contribute to variability in the intracellular pool of ribosomes (Ferretti et al, 2017; Genuth and Barna, 2018; Makino et al, 2021; Remacha et al, 1995; Kiparisov et al, 2005; Li and Wang, 2020; Taoka et al, 2016). While limited in scope, Hab1 may function as a mediator of tailored ribosomal degradation.

In conclusion, our analysis of Hab1 shows for the first time that cargo can be selectively loaded on the isolation membranes formed by bulk autophagy. The specificity of this loading depends on the presence of Atg8-PE on the isolation membrane. Our identification of such a receptor indicates that bulk autophagy is not a completely random process, and that cargo hitchhiking on bulk autophagy is apparent in the cell.

## Methods

### Yeast strains and media

Yeast strains and plasmids used in this study are listed in Table EV1 and EV2, respectively. Cells were grown in synthetic defined (SD) medium (0.17% Difco yeast nitrogen base without amino acids and ammonium sulfate, 0.5% ammonium sulfate, and 2% glucose, SDCA medium (0.17% Difco yeast nitrogen base without amino acids and ammonium sulfate, 0.5% casamino acids, 0.5% ammonium sulfate, and 2% glucose), or YPD medium (1% yeast extract, 2% peptone, and 2% glucose), as noted for each experiment. Cells were grown in liquid medium at 30 °C overnight until the OD$_{600}$ was ~1.0 before autophagy was induced, either by adding 200 ng/mL rapamycin or by shifting to a nitrogen-starvation (SD–N) medium (0.17% Difco yeast nitrogen base and 2% glucose). Reagents and primers used in this study are listed in Table EV3 and EV4, respectively.

### Yeast transformation

Yeast transformation was performed by a modified LiAc/PEG method (Schiestl and Gietz, 1989). Overnight cultures of 0.5 mL were transferred to 25 mL YPD medium and incubated for 4 h. Harvested cells were washed with LiSorb (10 mM Tris-HCl pH 8.0, 100 mM LiAc, 1 M D-sorbitol, 1 mM EDTA [ethylenediaminetetraacetic acid]) and resuspended in 150 μL of LiSorb. After adding 15 μL of 10 mg/mL single-stranded DNA, 50 μL of the cell was transferred to a new tube, and the remainder was stored at –80 °C. DNA fragments or plasmid to be transformed were supplemented before 300 μL of Li-PEG (40% w/v PEG3350, 10 mM Tris-HCl pH 8.0, 100 mM LiAc, 1 mM EDTA) was added to the suspension. After vortexing, the cells were incubated at room temperature for 30 min. 30 μL of dimethyl sulfoxide was added before treatment at 42 °C for 10 min as a heat shock. For auxotrophic markers, the cells

were suspended in sterile water and then plated on selection plates. For antibiotic resistance markers, the cells were suspended in YPD medium, incubated for 2 h, and then plated on a selection plate medium. The plate was incubated at 30 °C for 2–3 days, the colonies were picked up, and they were spread onto the next selection plate. For genomic manipulations, successfully transformed colonies were confirmed by PCR. For the construction of point mutants and N-terminal truncations of *HAB1*, a plasmid containing *HAB1* and its surrounding region was constructed and used. Mutations were introduced into the plasmid according to the PrimeSTAR® Mutagenesis Basal Kit instructions. Plasmid DNA was digested using restriction enzymes and inserted into the genome of *hab1Δ* cells by homologous recombination. Preparation of PCR fragments for transformation was performed according to established methods (Janke et al, 2004).

## Protein extraction from autophagic bodies

Autophagy-induced *atg15Δ* cells were collected by centrifugation at 3500 rpm for 3 min (R9A; Himac). The cells were washed with water and suspended in spheroplast buffer (50 mM Tris-HCl pH 7.5, 1.2 M sorbitol) containing 0.3% (v/v) 2-mercaptoethanol and 10 µg/ODU zymolyase 100 T, where $ODU = OD_{600}$ multiplied by the culture volume (mL). For cells cultured in SD medium, spheroplast buffer was prepared in SD medium to increase the efficiency of spheroplasting. Spheroplasts were then washed with ice-cold spheroplast buffer and disrupted by suspension in 16% Ficoll buffer (16% [w/v] Ficoll 400, 10 mM MES-Tris pH 6.9, 0.2 M sorbitol 100 µM $MgCl_2$) supplemented with cOmplete™ EDTA-free protease inhibitor cocktail. Spheroplasts were further subjected to 15 strokes in a Dounce homogenizer, and the resulting suspension was used as total lysate. For centrifugation, layers of 12% Ficoll buffer (12% Ficoll 400, 10 mM MES-Tris pH 6.9, 0.2 M sorbitol, 100 µM $MgCl_2$) and 0% Ficoll buffer (10 mM MES-Tris pH 6.9, 0.2 M sorbitol, 100 µM $MgCl_2$) were overlaid onto total lysate preparations in a centrifugation tube and centrifuged at 22,000 rpm for 1 h (SW28; Beckman Coulter). The resulting band between 12% and 0% layers was collected and resuspended in 16% Ficoll buffer. Following this, 7% Ficoll buffer (7% Ficoll 400, 10 mM MES-Tris pH 6.9, 0.2 M sorbitol, 100 µM $MgCl_2$) and 0% Ficoll buffer were overlaid onto the suspension, which was then subjected to centrifugation at 22,000 rpm for 30 min (SW40; Beckman Coulter). The middle layer between 7% and 0% Ficoll buffer was collected and again resuspended in 16% Ficoll buffer, overlaid with 7% and 0% Ficoll solutions, and subjected to a further round of centrifugation at 22,000 rpm for 30 min (SW40). The intermediate layer between 7% and 0% Ficoll buffer was collected and used as the vacuolar fraction. Vacuolar membranes were disrupted by passing samples through a 0.8 µm polycarbonate membrane filter and subjected to iodixanol density gradient (6–27%) centrifugation at 20,000 rpm for 90 min (SW40) in a buffer (30 mM MES-Tris pH 6.9, 0.7 M sorbitol, 0.1 M KCl, 0.5 mM $MgCl_2$). Gradients were fractionated using a Piston Gradient Fractionator (BIOCOMP). TCA (trichloroacetic acid) was added to each fraction to a final concentration of 10%, kept on ice for at least 10 min, and then centrifuged to remove the supernatant at 20,000 × g for 15 min. Ice-cold acetone was added to pellets, which were then centrifuged to remove the supernatant at 20,000 × g for 15 min. Dried pellets were suspended in sodium dodecyl sulfate (SDS) sample buffer (62.5 mM Tris-HCl pH 6.8, 2% w/v SDS, 10% glycerol, 5% 2-mercaptoethanol, 0.05 mg/mL bromophenol blue) and heated at 95 °C for 5 min.

## Peptide precipitation

To samples, methanol, chloroform, and ultrapure water at a ratio of 4:1, 1:1, and 4:1, were added respectively, with vortexing after each addition (all ratios to original volume). Samples were then centrifuged at 15,000 × g for 1 min at room temperature and the upper layer was discarded. Then, 3:1 volumes of methanol (relative to the original sample) were added, samples centrifuged at 15,000 × g for 2 min at room temperature, and the resulting supernatants were discarded. Pellets were dried using a centrifugal evaporator (TAITEC, VC-96R, 0062458-000) and dissolved in PTS solution (0.1 M Tris-HCl pH 9.0, 12 mM sodium deoxycholate, 12 mM sodium N-lauroyl sarcosinate). Protein concentrations were determined using the Pierce BCA protein assay kit and samples were diluted appropriately to 0.5 mg/mL. For every of 50 µL diluted samples, 0.5 µL of 1 M DTT in 50 mM $NH_4HCO_3$ was added before samples were incubated at room temperature for 30 min. Next, 2.5 µL of 50 mM iodoacetamide in 50 mM $NH_4HCO_3$ was added to samples, which were then incubated at room temperature for 30 min in the dark. After addition of 200 µL of 50 mM $NH_4HCO_3$, each sample was treated with Lys-C at 2% of the total protein for 3 h and then 2% trypsin for 18 h at 37 °C. 250 µL of ethyl acetate and then trifluoroacetic acid at a final concentration of 0.5% were added to the sample. Samples were vortexed and then centrifuged at 15,700 × g for 2 min at room temperature. Upper layers were discarded and the lower layers were dried using a centrifugal evaporator. Finally, peptide purification was performed using C18-StageTip (Rappsilber et al, 2007) (CDS, Empore™ Discs, C18).

## LC-MS/MS

Purified peptides were dried and dissolved in 25 µL 2% acetonitrile and 0.1% formic acid. An EASY-nLC 1000 (Thermo Scientific) equipped with a 125 mm × 75 µm C18 separation column (Nikkyo Technos, Co., Ltd.) was used for liquid chromatography. A Q-Exactive mass spectrometer (Thermo Scientific) was used for mass spectrometry. Separation was performed either with a 0–30% linear acetonitrile gradient in the presence of 0.1% formic acid for 80 min, followed by a 30–100% acetonitrile gradient for 2 min and a further 8 min in the presence of 100% acetonitrile (for Figs. 1C,D, EV3A, and EV4E), or with a 0–30% linear acetonitrile gradient in the presence of 0.1% formic acid for 35 min, followed by a 30–100% acetonitrile gradient for 10 min (for Fig. 5D). Data were acquired in data-dependent acquisition mode operated by Xcalibur 4.0 software (ThermoFisher Scientific). The setting of the data-dependent acquisition is as follows: the resolution was 70,000 for a full MS and 17,500 for $MS^2$; the AGC target was 1.0E6 for a full MS and 5.0E5 for $MS^2$; the maximum IT was 60 ms for both a full MS and $MS^2$; the scan range was 300 to 2000 *m/z* for a full MS; the top 10 signals were selected for $MS^2$; the isolation window was 3.0 *m/z* for $MS^2$.

## Data processing for LC-MS/MS

Acquired data were analyzed by Proteome Discoverer 2.4 (Thermo-Fisher Scientific) and the R programming language (Ver 4.0.2).

The fragmentation spectra were searched against the *S. cerevisiae* UniProt database (version Feb. 21, 2016) containing 6749 entries. The settings of Proteome Discoverer were as follows: maximum missed cleavages, 2; peptide length, 5–144; peptide mass tolerance, 10 ppm; fragment mass tolerance, ±0.02 Da. Meanwhile, oxidation, +15.995 Da (Met); propionamide, +71.037 Da (Cys); phospho, +79.966 Da (Ser, Thr, Tyr); GG dipeptide, +114.043 Da (Lys); Acetyl, +42.011 Da (N-terminus); Met-loss, –131.040 Da (N-terminus); and Met loss+Acetyl, –89.030 Da (N-terminus) were configured as variable modifications. Peptides were filtered to the 1% FDR target. For Fig. 1C,D, label-free quantification values in $MS^1$ in each sample were normalized and then compared to the control sample (corresponding to the autophagic body fraction in *atg2Δ* cells) and proteins that were more abundantly detected in the control sample were removed from the analysis. In order to retain proteins excluded from the autophagic body, removal was not performed for proteins that were not enriched in autophagic bodies. Proteins with extremely low abundance in total cell lysate ($<10^{-6}$) were excluded from this analysis due to the unreliability of quantification at these concentrations. Enrichment values for each protein in autophagic bodies were calculated as the ratio of abundance in autophagic bodies versus total cell lysate.

## Protein extraction for immunoblotting

1 or 3 OD unit of cells were obtained from cultures and kept on ice. Cell samples were treated with 10% TCA on ice for 10 min and centrifuged at $20,000 \times g$ for 15 min at 4 °C. The pellet was stored at −80 °C before use. The pellet was washed with ice-cold acetone and dried at room temperature, then resuspended in HU sample buffer (8 M urea, 0.2 M sodium phosphate pH 6.8, 5% w/v SDS, 10% glycerol, 0.1 M DTT, 0.1 M EDTA, 0.2 mg/mL bromophenol blue). Cells were disrupted with a half volume of 0.5-mm zirconia beads (Yasui Kikai, YZB05) using a FastPrep-24 (MP Biomedical) for 30 s at room temperature. Samples were incubated for 15 min at 65 °C with agitation. Finally, debris was removed by centrifugation at $20,000 \times g$ for 1 min before electrophoresis.

## Immunoblotting

Immunoblotting was performed as described previously (May et al, 2020; Iwama and Ohsumi, 2019). 12% polyacrylamide gels were used for GFP cleavage assays. Immunoblotting of the immunoprecipitated samples was performed on 13.5% polyacrylamide gels containing 6 M urea to separate Atg8-PE from Atg8. For Ponceau S staining, the membranes following transfer were washed with TBS-T and ultrapure water before staining with 0.1% Ponceau S in 5% acetic acid, and then washed with ultrapure water. Antibodies against GFP, HA, Pgk1, Pho8, Ape1, Atg8, Rpl23, and Hsp90 were used to detect relevant proteins. Horseradish peroxidase-conjugated secondary antibodies against rat IgG, mouse IgG, and rabbit IgG were used. All antibodies used in this study are listed in Table EV5. Chemiluminescence signals were obtained using a CCD camera system (Fusion FX; Vilber Lourmat). To calculate the degradation of GFP-fused proteins (%), GFP′ intensity was divided by the sum of GFP′ and GFP-fused full-length band intensities. The quantification of each band intensity was performed using ImageJ software (Fiji) (Schindelin et al, 2012).

## Fluorescence microscopy

Fluorescence microscopy was performed using an inverted fluorescence microscope (IX81; Olympus) or spinning disk confocal microscope (SpinSR10; Olympus). Fluorescence microscopy on the IX81 was performed as described previously (May et al, 2020), except that the IX81 was equipped with a CMOS camera (ORCA-Fusion; C15139-20U; Hamamatsu Photonics). The SpinSR10 was equipped with a CMOS camera (ORCA-Flash4.0 V3, C13440-20CU; Hamamatsu Photonics) and a 100× objective lens (UPLAPO 100XOHR, NA/1.5; Olympus). For FM4-64 staining, cells were incubated with medium containing 2 μM FM4-64 (Molecular Probes) for 30 min. The cells were then washed with fresh medium, shifted to medium containing rapamycin, and incubated for 4 h. Images were acquired using cellSens software (Olympus) and processed in the Fiji implementation of ImageJ (Schindelin et al, 2012).

## ALP assay

The ALP assay was performed as described previously (Noda and Klionsky, 2008).

## Immunoprecipitation

50 OD unit of cells were harvested, frozen in liquid nitrogen, and stored at –80 °C. Cell pellets were then suspended in 1 mL of ice-cold IP-lysis buffer (50 mM Tris-HCl pH 7.5, 150 mM NaCl, 10% glycerol, 1 mM EDTA, 50 mM NaF, 1 mM PMSF, 1 mM DTT, 1x protease inhibitor cocktail w/o EDTA [Roche]). 2 g of 0.5-mm zirconia beads were then added to samples before disruption using a Multi-Beads Shocker (Yasui kikai). The bottom of the tube was pierced with a needle and lysates were collected by centrifugation into a new tube. The lysates were treated with 0.5% TritonX-100 and rotated for 10 min at 4 °C. Debris was removed by two rounds of centrifugation ($500 \times g$, 4 °C, 5 min each). At this point, 25 μL of lysate was collected as an input sample. The lysates were mixed with 10 μL of GFP-trap M beads (ChromoTek) equilibrated with wash buffer (IP-lysis buffer, 0.3% TritonX-100), and rotated gently for 1 h at 4 °C. The beads were washed with wash buffer five times, and then treated with 50 μL of SDS sample buffer (62.5 mM Tris-HCl pH 6.8, 2% [w/v] SDS, 10% glycerol, 5% 2-mercaptoethanol, 0.05 mg/mL bromophenol blue) for 5 min at 95 °C to elute bound proteins.

## Immunoprecipitation for LC-MS/MS

250 OD unit of cells were harvested from cultures, frozen in liquid nitrogen, and stored at –80 °C. The pellet was crushed in liquid nitrogen using a ceramic mortar and pestle. The pellet powder was dissolved in 2.5 mL of ice-cold IP-lysis buffer supplemented with 0.1x PhosSTOP (Roche). Debris was removed by two rounds of centrifugation ($500 \times g$, 4 °C, 5 min each). The lysates were then mixed with 25 μL of GFP-trap M beads (ChromoTek) equilibrated with IP-lysis buffer and rotated gently for 1 h at 4 °C. The beads were washed with IP-lysis buffer six times and then treated with 100 μL of SDS sample buffer for 5 min at 95 °C to elute bound proteins.

## Subcellular fractionation

The pellet of 50 ODU of cells was suspended in 1 mL of ice-cold lysis buffer (20 mM HEPES-KOH pH 7.4, 100 mM CH$_3$COOK, 2 mM (CH$_3$COO)$_2$Mg, 1 mM DTT, 1 mM PMSF, 1.25x protease inhibitor cocktail, RNasin°Plus (Promega) at 40 units/mL), to which 2 g of 0.5-mm zirconia beads (Yasui kikai) was added and disrupted using a Multi-Beads Shocker (Yasui kikai). The lysate was centrifugated at $500 \times g$ and 4 °C for 5 min and the supernatant was collected. Debris was removed by two rounds of centrifugation ($9000 \times g$, 10 min, 4 °C each). RNA concentrations were measured using a NanoVue (GE Healthcare); concentrations were then adjusted to 0.5 mg/mL by the addition of lysis buffer. 25 μL of samples was acquired as input, and 475 μL of the sample was centrifuged at $10,0000 \times g$ for 1 h at 4 °C. The supernatants obtained were used as S$_{100}$ samples and the pellets were dissolved in lysis buffer for P$_{100}$ samples. 2x SDS sample buffer was added to samples, which were then heated at 95 °C for 5 min.

## Sucrose density gradient centrifugation

Debris-free lysates containing 1 mg/mL RNA were prepared using the same procedure as for subcellular fractionation. For cross-linking, the lysate was treated with dithiobis (succinimidyl propionate) (DSP) at a final concentration of 2 mM for 30 min at 30 °C with subsequent quenching by addition of Tris-HCl pH 7.5 to a final concentration of 20 mM. Sucrose gradients (10–50% sucrose in 10 mM HEPES-KOH pH 7.4, 70 mM ammonium acetate, 4 mM magnesium acetate) prepared in ultracentrifuge tubes (Hitachi Koki) using a Gradient Master (BIOCOMP) were layered with lysates containing 500 μg RNA and then centrifuged in a P40ST rotor (Hitachi Koki) at $130,000 \times g$ for 12 h at 4 °C. Gradients were fractionated using Piston Gradient Fractionator (BIOCOMP). Absorbance at 254 nm was measured using a BIO-MINI-UV monitor (ATTO, AC-5200L). To each fraction, an equal volume of 2xSDS sample buffer was added and then incubated for 5 min at 95 °C for immunoblotting.

## Transmission electron microscopy

Transmission electron microscopy (TEM) was performed by Tokai Electron Microscopy. Samples were inserted between copper disks and frozen at –175 °C in liquid propane for rapid freezing. Frozen samples were substituted using 2% glutaraldehyde and 1% tannic acid in ethanol for 2 days at –80 °C. Samples were then held for 2 h at –20 °C and warmed to 4 °C for 2 h. Dehydration with ethanol was performed three times for 30 min each, followed by continuous dehydration with ethanol overnight. The sample was infiltrated twice with propylene oxide for 3 min each, put in a 1:1 mixture of propylene oxide and resin for 3 h, and then transferred to 100% resin overnight. The resin was polymerized for 48 h at 60 °C. The resin block was ultra-thin sectioned at 80 nm with a diamond knife using an ultramicrotome (ULTRACUT; Leica), and the section was placed on copper grids. The sample was stained with 2% uranyl acetate for 15 min and then washed with distilled water after which secondary staining was performed using lead stain solution (Sigma-Aldrich Co.) for 3 min. Images were captured by a transmission electron microscope (JEM-1400Plus; JEOL Ltd.) equipped with a CCD camera (EM-14830RUBY2; JEOL Ltd.) at an acceleration voltage of 100 kV. The frequency of ribosomal association with autophagic membranes was determined using randomized electron micrographs of each condition under blind conditions.

## Statistical analyses

Statistical analyses were performed using the R programming language (Ver 4.0.2). For comparisons between conditions, the unpaired two-sided Welch's t-test was used. Box plots are shown as median (middle bar) with 25th and 75th percentiles.

## Data availability

Mass spectrometry data of cargo analyses are deposited in the jPOST repository (https://repository.jpostdb.org), jPOST ID: JPST002423.

## Peer review information

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

## Acknowledgements

We would like to thank Drs. Nobuo N. Noda, Hitoshi Nakatogawa, Tetsuya Kotani, Akinori Yamasaki, Yui Jin, and Tomoko Kawamata for critical comments; Drs. Tatsuya Niwa, Shiho Makino, and Tomoko Kawamata for helpful suggestions on LC-MS/MS analysis, sucrose density gradient centrifugation and the isolation of autophagic bodies, respectively; Dr. Hideki Taguchi for helping sucrose density gradient centrifugation; and Drs. Yasuhiro Araki and Kai Johnsson for the kind gift of plasmids containing GFP-binding protein and NbALFA, respectively. We are grateful to all the members of Ohsumi Laboratory for helpful discussions, and to the Biomaterials Analysis Division, Tokyo Institute of Technology for DNA sequencing. This work was

supported by JSPS KAKENHI (16H06375 and 19H05708 to YOh, 19K16121 and 22K15101 to ET, 19K06634 to YOi).

## Author contributions

**Eigo Takeda**: Conceptualization; Data curation; Formal analysis; Funding acquisition; Validation; Investigation; Methodology; Writing—original draft; Writing—review and editing. **Takahiro Isoda**: Conceptualization; Validation; Investigation. **Sachiko Hosokawa**: Data curation; Validation; Investigation. **Yu Oikawa**: Funding acquisition; Investigation; Methodology. **Shukun Hotta-Ren**: Resources. **Alexander I May**: Writing—review and editing. **Yoshinori Ohsumi**: Conceptualization; Supervision; Funding acquisition; Methodology; Writing—review and editing.

## Disclosure and competing interests statement

The authors declare no competing interests.

# Expanded View Figures

**Figure EV1.  Fluorescence microscopy of Hab1-GFP (related Fig. 2B).**

(**A**) Fluorescence micrographs of WT or *atg1Δ* cells expressing Hab1-GFP and Vph1-2xmCherry. Scale bar, 5 μm. (**B**) Fluorescence micrographs of Hab1-GFP and mCherry-Atg8-expressing cells after treatment with rapamycin for 1 or 4 h. Arrowheads show colocalization. Scale bar, 5 μm. (**C**) Time-lapse imaging of cells expressing Hab1-GFP and mCherry-Atg8. The time after rapamycin addition at which each snapshot was obtained is noted in the upper left corner of each. Arrowheads indicate mCherry-Atg8 or Hab1-GFP puncta. Scale bar, 5 μm.

▶

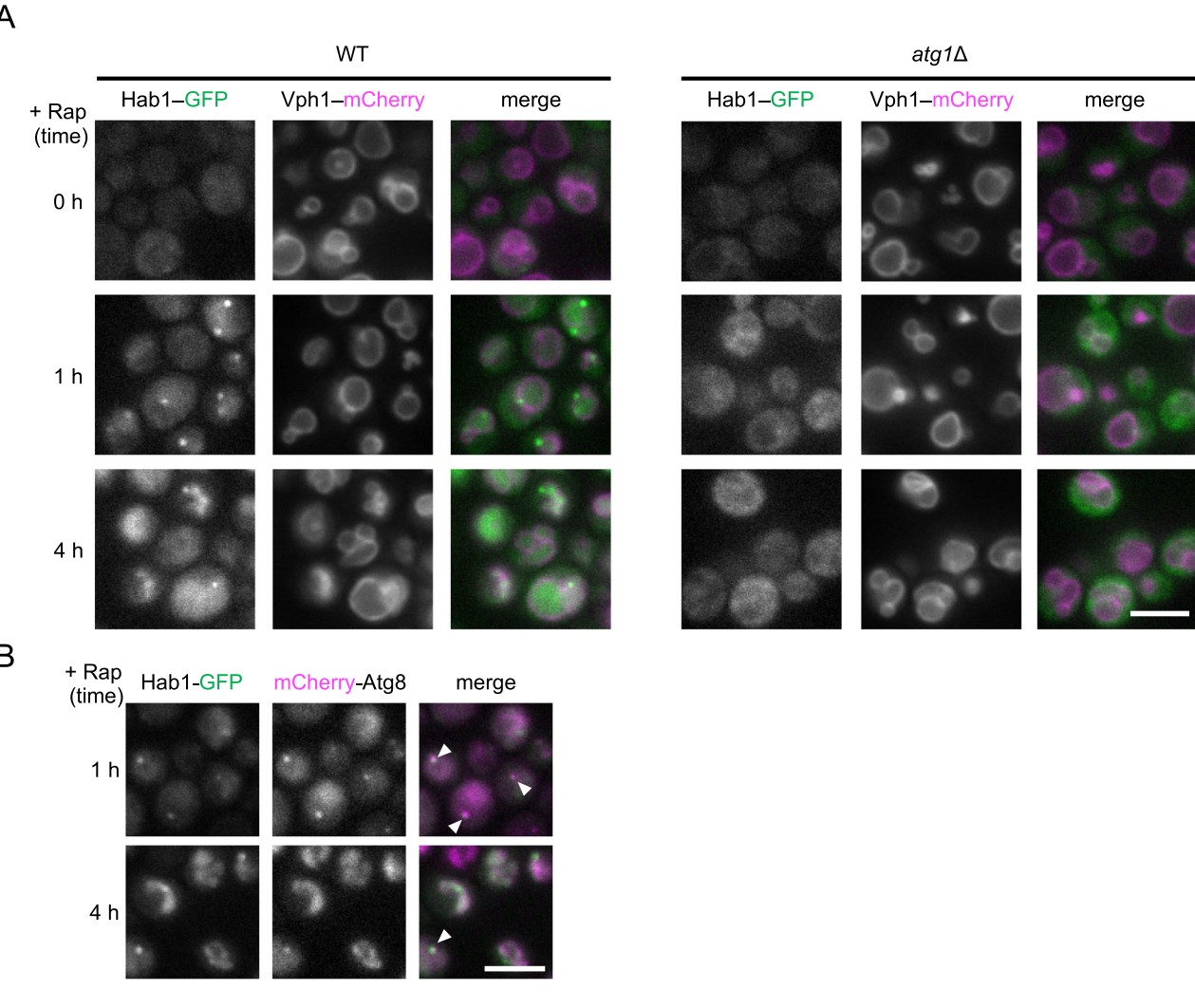

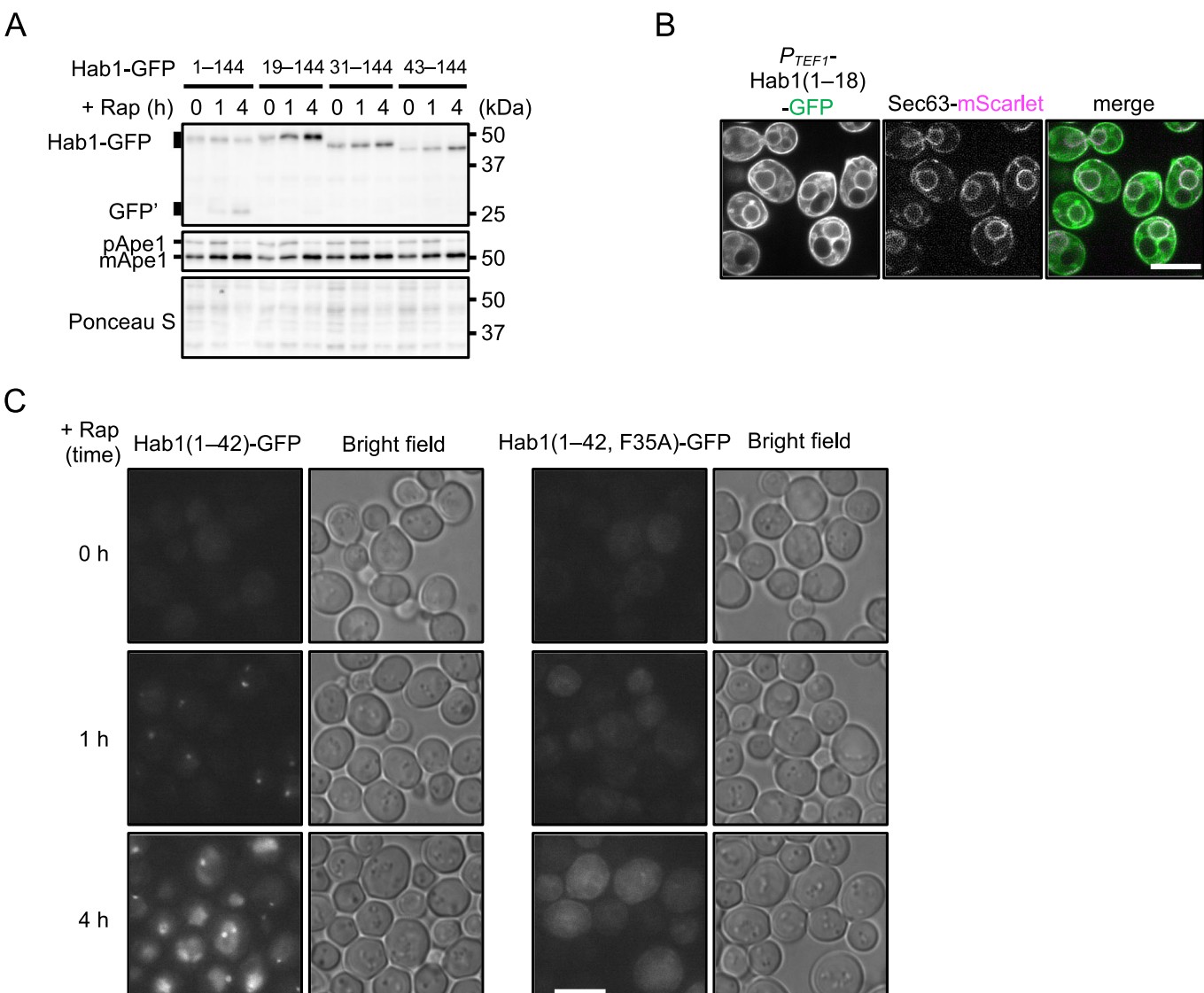

**Figure EV2. Hab1 N-terminal helix mutants (related to Fig. 3).**

(A) GFP cleavage of WT and N-terminal truncation mutants of Hab1. (B) Fluorescence micrographs of Hab1-GFP and Sec63-mScarlet (ER-membrane marker) observed by spinning disc confocal fluorescence microscopy. Scale bar, 5 μm. (C) Fluorescence micrographs of Hab1(1–42)-GFP-expressing cells. Scale bar, 5 μm.

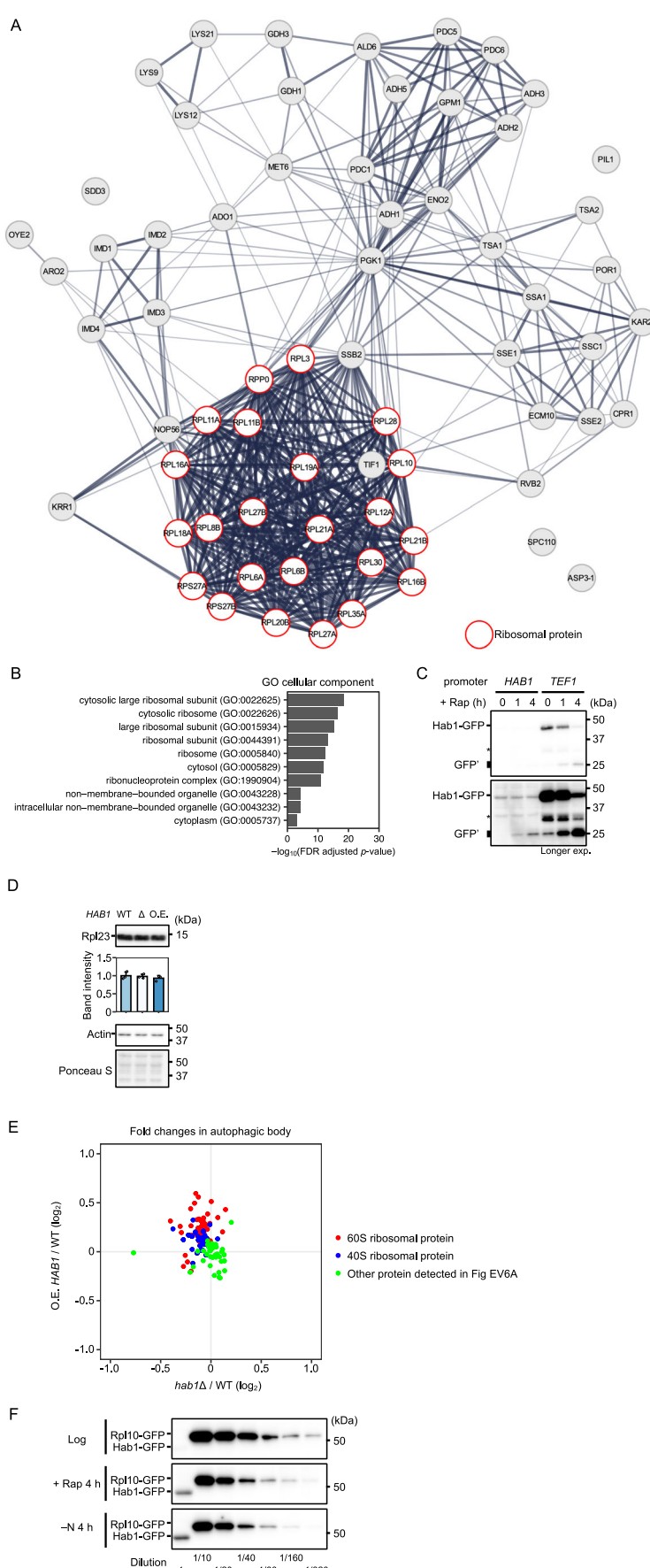

A

B GO cellular component

cytosolic large ribosomal subunit (GO:0022625)
cytosolic ribosome (GO:0022626)
large ribosomal subunit (GO:0015934)
ribosomal subunit (GO:0044391)
ribosome (GO:0005840)
cytosol (GO:0005829)
ribonucleoprotein complex (GO:1990904)
non−membrane−bounded organelle (GO:0043228)
intracellular non−membrane−bounded organelle (GO:0043232)
cytoplasm (GO:0005737)

−log₁₀(FDR adjusted p-value)

C

D

E Fold changes in autophagic body

● 60S ribosomal protein
● 40S ribosomal protein
● Other protein detected in Fig EV6A

F

**Figure EV3. Hab1 binds to ribosome (related to Fig. 5).**

(A) A network map of Hab1(43–144)-interacting proteins. Proteins immunoprecipitated with Hab1(43–144)-GFP were detected by LC-MS/MS. Detected Hab1(43–144)-GFP interacting proteins were analyzed using STRING (Szklarczyk et al, 2019) after control sample (GFP-immunoprecipitated) proteins were excluded. (B) GO enrichment analysis of Hab1-interacting proteins. Analysis was performed on the cellular component for all proteins identified in (A) using the GO Consortium (accessed August, 2022) (Ashburner et al, 2000). GOs with FDR-adjusted *p*-values lower than 0.01 are indicated. FDR was calculated using the one-way Fisher's exact test. (C) Confirmation of overexpression of Hab1 by placing *HAB1* under the control of the *TEF1* promoter. Cells expressing Hab1-GFP under the control of the *HAB1* or *TEF1* promoter were subjected to immunoblotting. Cells were cultured in SD media to the logarithmic growth phase and then treated with rapamycin to induce autophagy. (D) The effect of Hab1 on ribosome biosynthesis was evaluated by quantification of Rpl23. There was no detectable change in ribosomal protein levels in *atg15Δ*, *atg15Δ hab1Δ*, and *HAB1*-overexpressing *atg15Δ* cells in the logarithmic growth phase on YPD. Quantifications of band intensities are shown as mean ± SD. (*n* = 3); data are normalized to the WT band. (E) An analysis of non-ribosomal proteins in autophagic bodies detected in Fig. EV3A. Fold changes are shown as normalized label-free quantification of each 40 S or 60S ribosomal proteins in autophagic bodies isolated from *hab1Δ* (x-axis) or *HAB1*-overexpressing cells (y-axis) relative to that of WT cells. Green: non-ribosomal proteins detected in Fig. EV3A, red: 60S ribosomal proteins, blue: 40S ribosomal proteins. The same data set as shown in Fig. 5D was used. (F) Examination of abundance ratios of Hab1 to ribosomes by immunoblotting. Hab1-GFP- or Rpl10-GFP-expressing *atg2Δ* cells were cultured in YPD media to the logarithmic phase of growth and then treated with rapamycin or cultured in SD–N medium for 4 h. Cells were then harvested. For Rpl10-GFP-expressing cells, diluted samples were applied at the ratios described below each lane.

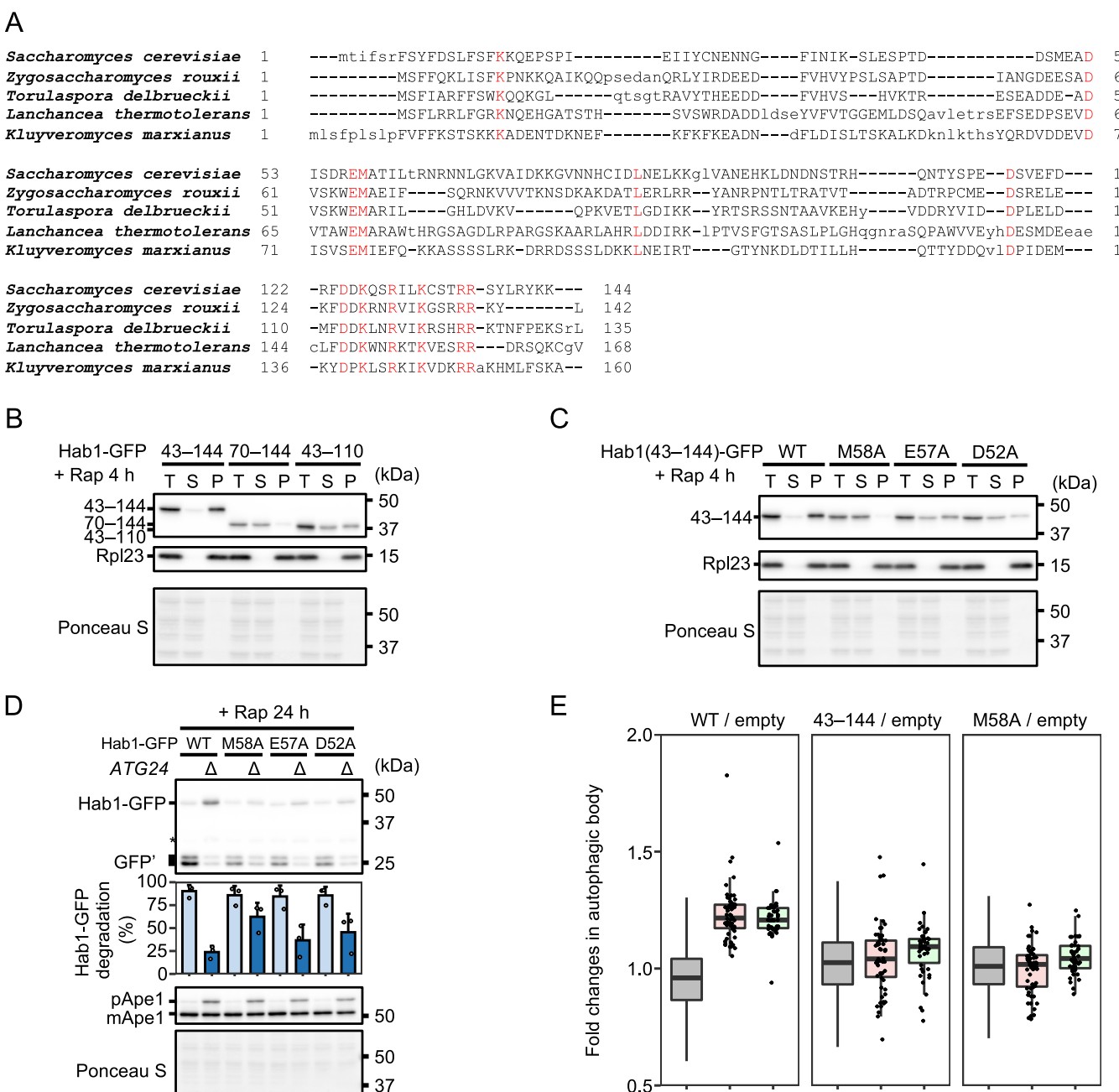

**Figure EV4. The N- and C-terminal regions of Hab1 are important for the Hab1-dependent delivery of ribosomes to vacuoles.**

(A) Hab1 orthologs of *Zygosaccharomyces rouxii*, *Torulaspora delbrueckii*, *Lanchancea thermotolerans,* and *Kluyveromyces marxianus* were identified based on sequence homology using PSI-BLAST searches. Amino acid sequences are shown. The alignment was performed using COBALT (https://www.ncbi.nlm.nih.gov/tools/cobalt/). Conserved residues are shown in red. (B) Immunoblotting of subcellular fractions. Lysates were obtained from cells expressing Hab1-GFP truncates. (C) Immunoblotting of subcellular fractions. Lysates were obtained from cells expressing Hab1-GFP point mutants. (D) GFP cleavage of Hab1 point mutants in WT and *atg24Δ* cells. The percentages of GFP' to total GFP (GFP' + full-length GFP) are shown as mean ± SD ($n = 3$). (E) Proteomic analyses of ribosomal proteins in autophagic bodies. Autophagic bodies isolated from *hab1Δ atg15Δ* cells harboring empty vector, *HAB1*(WT), *HAB1*(43–144) or *HAB1*(M58A) following 6 h rapamycin treatment were subjected to proteomic analyses. Fold changes are shown as normalized label-free quantifications of each 40S or 60S ribosomal protein in autophagic bodies. "Total" indicates all detected proteins. Box plots are shown as median (middle bar) with 25th and 75th percentiles and 1.5 × interquartile in whiskers.

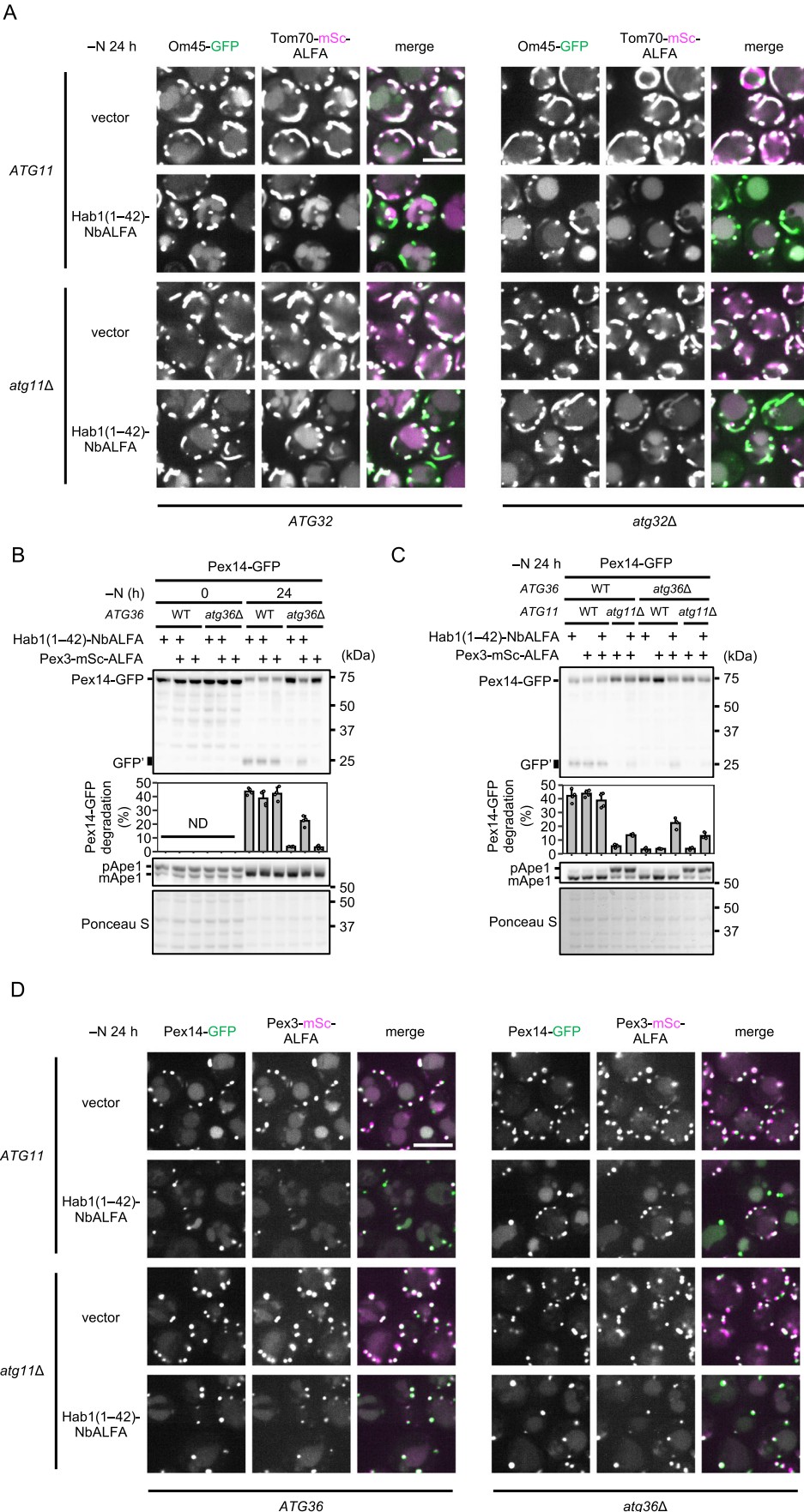

◄    **Figure EV5.  Binding to Hab1(1–42) allows receptor- and scaffold protein-independent delivery of organelles to vacuoles (related to Fig. 6).**

(A) Fluorescence microscope images of Hab1(1–42)-bound mitochondria showing delivery to the vacuole. Om45-GFP-expressing cells were observed 24 h after shifting to SD–N medium. Scale bar, 5 μm. (B) Rates of peroxisomal delivery to the vacuole were examined by GFP cleavage of Pex14-GFP. Hab1(1–42) was tethered to mitochondria using the ALFA-tag system. –N, nitrogen starvation; Percentages of GFP′ to total GFP (GFP′ + full-length GFP) are shown as mean ± SD ($n = 3$). ND, not determined. Blot data are representative of two independent experiments. (C) Rates of peroxisomal delivery to the vacuole were examined by GFP cleavage of Pex14-GFP. Percentages of GFP′ to total GFP (GFP′ + full-length GFP) are shown as mean ± SD ($n = 3$). A subset of data is derived from samples used for quantification in (B). Blot data are representative of three independent experiments. (D) Fluorescence micrographs during nitrogen starvation (related to Figs. EV5B and 5C). Scale bar, 5 μm.

                                   