## [Peer Review File · The EMBO Journal]

Receptor-mediated cargo hitchhiking on bulk autophagy

Eigo Takeda, Takahiro Isoda, Sachiko Hosokawa, Yu Oikawa, Shukun Ren, Alexander May, and Yoshinori Ohsumi

Corresponding author(s): Yoshinori Ohsumi (ohsumi.y.aa@m.titech.ac.jp) , Eigo Takeda (takeda.e.aa@m.titech.ac.jp) , Yoshinori Ohsumi (ohsumi.y.aa@m.titech.ac.jp)

Review Timeline:

Submission Date:	22nd Aug 23
Editorial Decision:	9th Oct 23
Revision Received:	10th Jan 24
Editorial Decision:	12th Feb 24
Revision Received:	24th Feb 24
Accepted:	28th Feb 24

Editor: *Cornelius Schneider*

Transaction Report:

Dear Prof. Ohsumi,

Thank you for submitting your manuscript for consideration by the EMBO Journal. It has now been seen by three referees whose comments are shown below.

As you will see from the reports, the reviewers appreciate the work, while also indicating a number of constructive points that would need to be addressed before acceptance here. From my side, I find the reviewer comments reasonable and constructive. Therefore, based on these positive assessments, I would like to invite you to address the issues raised by the reviewers in a revised manuscript.

I would be happy to discuss the revision in more detail via email or phone/videoconferencing.

We generally allow three months as standard revision time. As a matter of policy, competing manuscripts published during this period will not negatively impact on our assessment of the conceptual advance presented by your study. However, please contact me as soon as possible upon publication of any related work to discuss the appropriate course of action. Should you foresee a problem in meeting this three-month deadline, please contact us to arrange an extension.

When preparing your letter of response to the referees' comments, please bear in mind that this will form part of the Review Process File and will therefore be available online to the community. For more details on our Transparent Editorial Process, please visit our website: <https://www.embopress.org/page/journal/14602075/authorguide#transparentprocess>. Please also see the attached instructions for further guidelines on preparation of the revised manuscript.

Please feel free to contact me if you have any further questions regarding the revision. Thank you for the opportunity to consider your work for publication, and I look forward to your revision.

With best regards,

Cornelius Schneider

Cornelius Schneider, PhD
Editor
The EMBO Journal
c.schneider@embojournal.org

We realize that it is difficult to revise to a specific deadline. In the interest of protecting the conceptual advance provided by the work, we recommend a revision within 3 months (7th Jan 2024). Please discuss the revision progress ahead of this time with the editor if you require more time to complete the revisions. Use the link below to submit your revision:

Referee #1:

In this very interesting paper Takeda et al. do a proteomic screen in yeast to characterize autophagic substrates and identifies a small 144 amino acid long protein they call Hab1 that they find to interact with the forming autophagosomes (phagophores) via a 42 amino acid N-terminal region harboring an amphipathic helix and an AIM/LIR motif that binds to lipidated Atg8. These two features in this short 42 amino acid region are sufficient for preferential or selective degradation of Hab1 by autophagy. This region can be fused to another yeast protein, or even to mitochondrial or peroxisomal membrane proteins mediating the selective delivery of the protein or the respective organelles to the vacuole. Hab1 itself, or the fused components, are degraded by autophagy without need for the scaffold protein for selective autophagy in yeast, Atg11. The C-terminal part of Hab1 from amino acid 43 to 144 has the ability to bind to ribosomes. The authors present data to suggest that Hab1 has a role in delivering ribosomes to the vacuole via autophagy. This is not a major ribophagy receptor, but it has a role. The authors propose that Hab1 acts in a novel mode by binding directly to the forming autophagosomes during bulk autophagy.

This is a very interesting paper, with excellent experimental strategies and well controlled experiments where the conclusions made are generally well supported by the data shown.

1. In Figs 4A-C it would be good to see long exposures also of the IP blots since the WT control band in the IPs is quite weak.

2. In relation to Fig 5E the authors write: "Ribosomes, which are visualized as dots of high electron density, were enriched on the inner surface of autophagic body membranes in Hab1-overexpressing cells (Fig 5E, arrowheads)."

I was wondering how certain can the authors be that the black dots at the inner surface of the membrane in the EM images are ribosomes? It would be nice if the authors could verify this by immune-EM.

3. The title of the paper is "A novel form of selective autophagy by cargo hitchhiking on bulk autophagy". The implication is that bulk autophagy is not completely unselective. An autophagy receptor like Hab1 can interact with the isolation membrane and get degraded itself and also take cargo with it like a subset of ribosomes. I can understand why this title was chosen, but it is not cargo as such which is hitchhiking, it is the receptor, that may or may not, have bound cargo. I will not insist on a change of title, but a more precise title would be something like: "Receptor-mediated selective autophagy hitchhiking on bulk autophagy".

4. Do the authors think it may be likely that Hab1 can have other substrates than ribosomes?

Are there other interactors known for the C-terminal 43-144 part of the molecule?

5. Is it likely, that there is a mammalian homolog or protein with similar function?

Referee #2:

Summary

The authors describe a novel form of preferential degradation via autophagy independent of the scaffold protein Atg11. Via mass spectrometry analysis of isolated autophagic bodies they identified a highly abundant yet uncharacterized protein that they named Hab1. Hab1 contains an Atg8 interacting motif (AIM) and an amphipathic helix at the N-terminus. Both are required for protein sequestration by autophagy. The authors propose that Hab1 interacts directly with Atg8-PE on the membrane via an amphipathic helix and an AIM. This enables Hab1 to act as an adaptor protein for ribosomal degradation as it is shown to bind to ribosomes via its C-terminus. Additionally, they show that expression of the N-terminal part of Hab1 is sufficient to direct preferential degradation of targeted organelles.

Overall, the paper is well written and the results novel and interesting as this is the first report of a protein module that exploits membrane and Atg8 binding to selectively recruit cargo to starvation induced autophagy. It is expected that more of these modules become known in the future. As such it is interesting for the wider autophagy community. A few points should be

addressed in order to further improve the manuscript.

Individual comments:

1. The frequent lack of Western Blot quantifications prevents the reader from evaluating the presence of mild phenotypes. One example is Figure 2D where the overall degradation is low and is difficult to assess from one blot. Other examples are the blots for the IPs shown in Figure 4 as well as the blots in Figure 5A, B. These quantifications should be based on at least 3 experiments.
2. The relationship between the amphipathic helix and the AIM mutant is not entirely clear. How do 1-42 and its AIM mutant localize?
3. Why are the IP experiments performed with the truncated forms and not the full-length Hap1? In Figure 4C the lipidation status in the input between wt and mutants is quite different. Can it have an effect on the IP? Furthermore, according to the methods the IPs were conducted in the presence of detergent, which should disrupt the membrane. How can Hap1 1-42 still have a preference for lipidated Atg8 under these conditions?
4. The analysis of the mass spec data should be described better. For example, what does the x-axis in Figure 1C refer to. Also, the analysis resulting in Figure EV5 should be elaborated on and a volcano plot should be shown. Furthermore, the mass spec data should be deposited in a depository and made publicly available after publication.
5. Figure 5F: What does the term "rate" refer to in the y-axis. Do the authors mean frequency or density and how was this value derived?
6. Hab1 overexpressing cells show more ribosomes in the autophagic bodies. Is the expression level of ribosomal proteins the same at t0? Is it possible that Hab1 overexpression induces ribosome biogenesis? The concern also applies to the EM analysis

Referee #3:

Autophagy is traditionally viewed as either exclusively selective for specific cargo to be eliminated or strictly non-selective, wherein bulk cytoplasm is engulfed randomly. The ability of non-selective autophagy to nevertheless show a preference for specific cargo is a long-standing issue that remains elusive. In their submitted manuscript, Takeda et al. profile autophagic bodies that accumulate upon induction of nonselective autophagy by rapamycin and identify enrichment of the novel protein Hab1, which is degraded in an autophagy-dependent manner, which specifically requires Atg24. The authors elegantly show that the N-terminal 42 amino-acids of Hab1 act as a degron that may forcibly sequester degron-attached cargo independently of known selectivity factors, owing to the combined interaction of this degron with membranes as well as with lipidated Atg8. This mechanism appears to bear physiological relevance, as expression of Hab1 is required and sufficient for efficient autophagic sequestration of ribosomes, which in turn are found to interact with its C-terminal 43-144 fragment. A novel mode of selectivity within bulk autophagy is thus offered but requires additional analysis as follows:

1. Fig. 1C The complete list of cargo enrichment indices should be provided for transparency. A pulse-chase assay should follow the elimination kinetics of specific protein cohorts over the course of autophagy for Hab1 versus other model cargos, namely Pgk1, Ald6, and 40S and 60S ribosomal protein, in wildtype versus autophagy knockout strains to establish Hab1 as preferential cargo convincingly.
2. The sequestration of peroxisomal and mitochondrial fragments by artificial interaction with Hab1 in Atg11, Atg36/32 double knockout cells (Fig. 6E, EV6, EV7) would benefit from visualization by electron microscopy for both phagophores and autophagic bodies, preferably with the labeling of Hab1 to establish its direct physical contribution to artificial sequestration of large cargo. While Atg24 is shown to be dispensable for the degradation of the Hab1 degron itself (Fig. 5A), the requirement of Atg24 for degron-mediated sequestration of organelles should also be evaluated.
3. Hab1 43-144 exclusively interacts with the large - but not small - ribosomal particle (Fig. EV5A), yet migrates with the 80S particle (Fig. 5C) and facilitates sequestration of both small and large ribosomal subunits (Fig. 5D). To establish a direct physiological role for Hab1-mediated autophagic sequestration, a specific ribosomal interaction partner should be identified and assayed for interaction with the full-length Hab1 under both growth and autophagy-inducing conditions. Moreover, the Hab1-dependent autophagic proteome should be assayed not only for ribosomal proteins (Fig. 5D) but in a more comprehensive manner as in Fig. 1C, for Hab1 knockout vs. 1-144, 1-42 and 43-144 variants - to demonstrate the combined contribution of these domains to Hab1-mediated autophagy of specific cargo proteins.
4. The requirement of Atg24 for efficient degradation of full-length Hab1 (Fig. 5C) is attributed to the latter's interaction with ribosomes. However, other large cargos (Fas1, proteasomal particles) also require Atg24 for autophagic elimination. It is, therefore, unclear whether endogenous elimination of full-length Hab1 is mediated by ribosomes, other Atg24-dependent cargo, or another Atg24-dependent mechanism. Therefore, the precise localization of full-length Hab1 on wildtype and Atg24-deficient phagophores and its colocalization and physical interaction with Atg24 should be experimentally addressed.

Referee #1:

*In this very interesting paper Takeda et al. do a proteomic screen in yeast to*
*characterize autophagic substrates and identifies a small 144 amino acid long*
*protein they call Hab1 that they find to interact with the forming*
*autophagosomes (phagophores) via a 42 amino acid N-terminal region*
*harboring an amphipathic helix and an AIM/LIR motif that binds to lipidated*
*Atg8. These two features in this short 42 amino acid region are sufficient for*
*preferential or selective degradation of Hab1 by autophagy. This region can be*
*fused to another yeast protein, or even to mitochondrial or peroxisomal*
*membrane proteins mediating the selective delivery of the protein or the*
*respective organelles to the vacuole. Hab1 itself, or the fused components, are*
*degraded by autophagy without need for the scaffold protein for selective*
*autophagy in yeast, Atg11. The C-terminal part of Hab1 from amino acid 43 to*
*144 has the ability to bind to ribosomes. The authors present data to suggest that*
*Hab1 has a role in delivering ribosomes to the vacuole via autophagy. This is not*
*a major ribophagy receptor, but it has a role. The authors propose that Hab1*
*acts in a novel mode by binding directly to the forming autophagosomes during*
*bulk autophagy.*
*This is a very interesting paper, with excellent experimental strategies and well*
*controlled experiments where the conclusions made are generally well supported*
*by the data shown.*

Thank you for the very positive evaluation of our manuscript and
valuable suggestions.

*1. In Figs 4A-C it would be good to see long exposures also of the IP blots since*
*the WT control band in the IPs is quite weak.*

Thank you for this suggestion. In this revision we provide western blot
data showing a longer exposure time in Figs 4A-C.

*2. In relation to Fig 5E the authors write: "Ribosomes, which are visualized as*
*dots of high electron density, were enriched on the inner surface of autophagic*

body membranes in Hab1-overexpressing cells (Fig 5E, arrowheads)."
I was wondering how certain can the authors be that the black dots at the inner
surface of the membrane in the EM images are ribosomes? It would be nice if the
authors could verify this by immune-EM.

Ribosomes, which were first observed using electron microscopy by
George E. Palade and colleagues in the 1950s, have been observed in the
cytoplasm of eukaryotic cells as 25 to 30 nm diameter particles of high electron
density for many years. We also used EM in our early studies of autophagy to
show the delivery of ribosomes to the vacuole by observing such ribosomal
particles within autophagic bodies (Takeshige *et al.*, 1992; PMID: 1400575).

We understand that the ribosomes observed in the original version of
the manuscript may not be immediately apparent to readers who are not familiar
with this background. Accordingly, we have provided new, higher-quality
images in this revision and added an image of rough ER to provide context for
these data. The images show that ribosomes associated with rough ER are
morphologically indistinguishable from the particles we observe clustered in the
vicinity of autophagic body membranes. While we did not perform the suggested
experiment (immunoelectron microscopic validation of ribosomes) due to
numerous technical difficulties and the general acceptance of such electron dense
structures as ribosomes in the literature, we trust that the referee will be reassured
that these structures are indeed ribosomes.

*3. The title of the paper is "A novel form of selective autophagy by cargo*
*hitchhiking on bulk autophagy". The implication is that bulk autophagy is not*
*completely unselective. An autophagy receptor like Hab1 can interact with the*
*isolation membrane and get degraded itself and also take cargo with it like a*
*subset of ribosomes. I can understand why this title was chosen, but it is not*
*cargo as such which is hitchhiking, it is the receptor, that may or may not, have*
*bound cargo. I will not insist on a change of title, but a more precise title would*
*be something like: "Receptor-mediated selective autophagy hitchhiking on bulk*
*autophagy".*

We appreciate this suggestion from referee #1. We have changed the title to
"Receptor-mediated cargo hitchhiking on bulk autophagy".

*4. Do the authors think it may be likely that Hab1 can have other substrates than*
*ribosomes?*

*Are there other interactors known for the C-terminal 43-144 part of the*
*molecule?*

We interpret the data that we have collected as indicating that
ribosomes are the sole substrate of Hab1. To explore this question further, we
reanalyzed Hab1 cargo (Fig 5F) to find other proteins in the IP/MS analysis of
Hab1(43-144) (Fig EV6A in the revised manuscript). However, in contrast to
ribosomal proteins, we were unable to identify other proteins that were
consistently delivered to the vacuole in a Hab1-dependent manner (Fig EV6E in
the revised manuscript). We have added a comment discussing these results to
the revised manuscript (Line 224, in the revised manuscript).

*5. Is it likely, that there is a mammalian homolog or protein with similar*
*function?*

We could not find any mammalian homolog with amino acid sequence
similarity to Hab1 using standard bioinformatic approaches. However, we did
identify homologs in yeast species related to *Saccharomyces cerevisiae*. We have
added information about these homologs to the manuscript (Fig EV7A, line 251,
in the revised manuscript). As mentioned in the discussion, there may be
functional homologs in mammals and other organisms that bind specifically to
Atg8-PE and are subsequently delivered to lysosomes in a bulk
autophagy-dependent manner. By describing this novel form of cargo hitchhiking,
we anticipate that such functional homologs will be identified by our colleagues
in the mammalian research community.

Referee #2:

*The authors describe a novel form of preferential degradation via autophagy*
*independent of the scaffold protein Atg11. Via mass spectrometry analysis of*
*isolated autophagic bodies they identified a highly abundant yet uncharacterized*
*protein that they named Hab1. Hab1 contains an Atg8 interacting motif (AIM)*
*and an amphipathic helix at the N-terminus. Both are required for protein*
*sequestration by autophagy. The authors propose that Hab1 interacts directly*
*with Atg8-PE on the membrane via an amphipathic helix and an AIM. This*
*enables Hap1 to act as an adaptor protein for ribosomal degradation as it is*
*shown to bind to ribosomes via its C-terminus. Additionally, they show that*
*expression of the N-terminal part of Hab1 is sufficient to direct preferential*
*degradation of targeted organelles.*

*Overall, the paper is well written and the results novel and interesting as this is*
*the first report of a protein module that exploits membrane and Atg8 binding to*
*selectively recruit cargo to starvation induced autophagy. It is expected that*
*more of these modules become known in the future. As such it is interesting for*
*the wider autophagy community. A few points should be addressed in order to*
*further improve the manuscript.*

Thank you for very positive review and insightful remarks on our
manuscript.

*1. The frequent lack of Western Blot quantifications prevents the reader from*
*evaluating the presence of mild phenotypes. One example is Figure 2D where the*
*overall degradation is low and is difficult to assess from one blot. Other*
*examples are the blots for the IPs shown in Figure 4 as well as the blots in*
*Figure 5A, B. These quantifications should be based on at least 3 experiments.*

We have added quantifications of three independent experiments for all
figures indicated by referee #3 in the revised manuscript.

*2. The relationship between the amphipathic helix and the AIM mutant is not*
*entirely clear. How do 1-42 and its AIM mutant localize?*

To address this comment, we performed confocal fluorescence
microscopy of Hab1(1-42)-GFP and Hab1(1-42, F35A)-GFP expressing cells
following rapamycin treatment. The results of these experiments are shown in
Fig EV3C. In contrast to wild-type cells, Hab1(1-42, F35A)-GFP expressing
cells show neither Hab1 assemblies in the cytoplasm nor enrichment of Hab1 in
the vacuoles under these conditions.

*3. Why are the IP experiments performed with the truncated forms and not the*
*full-length Hap1? In Figure 4C the lipidation status in the input between wt and*
*mutants is quite different. Can it have an effect on the IP? Furthermore,*
*according to the methods the IPs were conducted in the presence of detergent,*
*which should disrupt the membrane. How can Hap1 1-42 still have a preference*
*for lipidated Atg8 under these conditions?*

Thank you for this comment. As we found in Fig 3B that Hab1(1-42)
alone is preferentially degraded, we used the truncated form of Hab1 in the
experiments shown in Fig 4 to determine the minimum conditions required for
degradation of Hab1. Separately, we confirmed that full-length Hab1 binds to
Atg8-PE, a result that has been added as Fig EV4 in the revised version of the
manuscript.

As referee #2 notes, the Atg8-PE ratio increases when Hab1, which
binds to Atg8-PE, is expressed. However, these changes are not large (Figures
4A, 4B, and 4C), and we do not believe such changes affect the IP results shown
in the paper.

We added detergent in IP experiments in an attempt to optimize the
recovery of Atg8-PE. While the presence of a detergent results in the disruption
of many membranes in lysate samples, some phospholipids may remain. Our
data suggest that Hab1 is still able to bind Atg8-PE and/or other phospholipids
that remain intact in the vicinity of Atg8-PE in the presence of detergent.

*4. The analysis of the mass spec data should be described better. For example,*
*what does the x-axis in Figure 1C refer to. Also, the analysis resulting in Figure*

*EV5 should be elaborated on and a volcano plot should be shown. Furthermore,*
*the mass spec data should be deposited in a depository and made publicly*
*available after publication.*

We are sorry for omitting these important points and thank the reviewer
for helping to improve the clarity of our data presentation. We have added
explanations to the legends and methods (Figs 1C and D) to address this
comment. Regarding Fig EV6A (in the revised manuscript), our data set cannot
be presented as a volcano plot because it was performed as a single-shot analysis,
and many protein signals detected in the Hab1(43-144)-GFP sample were not
detected at all in the GFP-expressing control sample. Since the purpose of this
analysis was to search for candidate interaction partners of Hab1(43-144) rather
than a detailed quantitative analysis of protein enrichment, we believe that the
current presentation of the data is sufficient.

We will deposit the raw mass spec data and methods (related to Figs 1C
and 1D) in the JPOST repository
(<https://repository.jpostdb.org/entry/JPST002423>) following publication of this
manuscript.

*5. Figure 5F: What does the term "rate" refer to in the y-axis. Do the authors*
*mean frequency or density and how was this value derived?*

The y-axis in Fig 5F was confusing, so we changed the axis label to
“frequency”. Also, we added detailed explanations of the experimental approach
to the legend and the methods section of the paper (Please see the section
“Transmission electron microscopy”).

*6. Hab1 overexpressing cells show more ribosomes in the autophagic bodies. Is*
*the expression level of ribosomal proteins the same at t0? Is it possible that Hab1*
*overexpression induces ribosome biogenesis? The concern also applies to the*
*EM analysis*

We conducted an evaluation of the effect of Hab1 on ribosome

biosynthesis, the results of which are presented in Fig EV6D. Based on this
analysis we can confirm that there was no significant change in ribosomal protein
levels among the strains assessed in Figs 5D, 5E, and 5F.

Referee #3:

*Autophagy is traditionally viewed as either exclusively selective for specific*
*cargo to be eliminated or strictly non-selective, wherein bulk cytoplasm is*
*engulfed randomly. The ability of non-selective autophagy to nevertheless show a*
*preference for specific cargo is a long-standing issue that remains elusive. In*
*their submitted manuscript, Takeda et al. profile autophagic bodies that*
*accumulate upon induction of nonselective autophagy by rapamycin and identify*
*enrichment of the novel protein Hab1, which is degraded in an*
*autophagy-dependent manner, which specifically requires Atg24. The authors*
*elegantly show that the N-terminal 42 amino-acids of Hab1 act as a degron that*
*may forcibly sequester degron-attached cargo independently of known selectivity*
*factors, owing to the combined interaction of this degron with membranes as*
*well as with lipidated Atg8. This mechanism appears to bear physiological*
*relevance, as expression of Hab1 is required and sufficient for efficient*
*autophagic sequestration of ribosomes, which in turn are found to interact with*
*its C-terminal 43-144 fragment. A novel mode of selectivity within bulk*
*autophagy is thus offered but requires additional analysis as follows:*

Thank you for the positive evaluation of our manuscript.

*1. Fig. 1C The complete list of cargo enrichment indices should be provided for*
*transparency. A pulse-chase assay should follow the elimination kinetics of*
*specific protein cohorts over the course of autophagy for Hab1 versus other*
*model cargos, namely Pgk1, Ald6, and 40S and 60S ribosomal protein, in*
*wildtype versus autophagy knockout strains to establish Hab1 as preferential*
*cargo convincingly.*

The MS data shown in Fig 1C is the first attempt at a comprehensive
analysis of the protein content of autophagic bodies. Our intention in these
experiments was to identify proteins that are degraded with very high efficiency
by autophagy, which led to our identification of Hab1. As Hab1 is an
uncharacterized protein with an exciting role in ribosomal degradation, we chose
to focus on this protein in this manuscript rather than perform a detailed

quantitative analysis of protein enrichment in autophagic bodies, which would
require a whole separate study to properly describe. While we confirmed the
preferential degradation of Hab1 in Fig 2 using well-established quantitative
assays (GFP cleavage assay and fluorescence microscopy analyses), we are not
confident that the MS data shown in Fig 1C are sufficiently exhaustive to
accurately quantify the selectivity of autophagy for every protein. We agree that
a detailed analysis of autophagy cargo by mass spectrometry is important, but it
would require further extensive analyses that we feel are not the main purpose of
this paper. In the interests of transparency, we have provided source data of the
plot of Figs 1C and 1D, and will provide raw data for this experiment by
depositing it to the JPOST repository
(<https://repository.jpostdb.org/entry/JPST002423>) following acceptance of this
paper.

Further, we respectfully disagree that pulse-chase experiments are
necessary in this study. We did not set out to compare the relative selectivity of
degradation among a range of autophagy cargoes; rather, our aim is to elucidate
the mechanism by which Hab1 is preferentially degraded. We believe that we
have presented a comprehensive characterization of the role of Hab1 in
autophagy, and that our analyses provide a significant advance in our
understanding of the selectivity of autophagy.

*2. The sequestration of peroxisomal and mitochondrial fragments by artificial*
*interaction with Hab1 in Atg11, Atg36/32 double knockout cells (Fig. 6E, EV6,*
*EV7) would benefit from visualization by electron microscopy for both*
*phagophores and autophagic bodies, preferably with the labeling of Hab1 to*
*establish its direct physical contribution to artificial sequestration of large cargo.*
*While Atg24 is shown to be dispensable for the degradation of the Hab1 degran*
*itself (Fig. 5A), the requirement of Atg24 for degran-mediated sequestration of*
*organelles should also be evaluated.*

While we had initially attempted immuno-EM to detect Hab1, the
frequency of the signal detection was low, even upon Hab1 overexpression,
suggesting technical issues in the antibody-epitope interaction. However, we

determined that even if immuno-EM experiments were successful, such data
would not provide significant new insights into the mechanism of artificially
induced Hab1-mediated organelle degradation and therefore did not pursue this
angle any further. Further, extensive and protracted optimization of conditions
for immuno-EM allowing detection of Hab1 would likely only show that Hab1 is
observed between autophagic membranes and the artificial cargo. This result
would not be particularly informative given the presence of Atg8-PE on the
membrane and the affinity of Hab1 for this molecule (Figs 5E and F). We
therefore feel that the suggested EM experiments would only reproduce Figs 6E,
EV6 or EV8 (EV7 in the previous version) by other means and would not justify
the extensive additional work required to optimize experimental conditions.

We found that Atg24 is only partially required for mitochondrial
degradation by the intrinsic Atg32-mediated pathway, and that the requirement
for Atg24 is even more marginal when mitochondria are degraded in an artificial,
Hab1-mediated manner (See the attached Panel A below these responses). These
observations may be informative in terms of Atg24 and selective organelle-phagy
research, but we think that these results and a discussion of their implications for
these fields are beyond the scope of the current paper. We have therefore chosen
not to include these data in our revised manuscript.

*3. Hab1 43-144 exclusively interacts with the large - but not small - ribosomal*
*particle (Fig. EV5A), yet migrates with the 80S particle (Fig. 5C) and facilitates*
*sequestration of both small and large ribosomal subunits (Fig. 5D). To establish*
*a direct physiological role for Hab1-mediated autophagic sequestration, a*
*specific ribosomal interaction partner should be identified and assayed for*
*interaction with the full-length Hab1 under both growth and autophagy-inducing*
*conditions. Moreover, the Hab1-dependent autophagic proteome should be*
*assayed not only for ribosomal proteins (Fig. 5D) but in a more comprehensive*
*manner as in Fig. 1C, for Hab1 knockout vs. 1-144, 1-42 and 43-144 variants -*
*to demonstrate the combined contribution of these domains to Hab1-mediated*
*autophagy of specific cargo proteins.*

We examined the interaction between Hab1 and ribosomes under both

growing and autophagy-inducing conditions. The native expression of Hab1-GFP
was low under growing conditions, and we found that interactions with
ribosomal proteins were not easy to determine. Therefore, the same experiment
was performed using cells in which Hab1-GFP was expressed under the control
of the *TEF1* promoter (an overexpression condition). We found that the
interaction between Hab1 and ribosomes did not significantly change under these
conditions (See attached panel B). We further note that Hab1 expression is
highly upregulated under autophagy-inducing conditions (Figs 2C and EV2).
These results indicate that Hab1-mediated ribosome delivery may be regulated
by the expression of Hab1 rather than the binding affinity of this protein.

The identification of the region of the ribosome that interacts with Hab1
is an interesting question, but this would require a large amount of additional
experiments that would be technically very demanding. Even if the binding site
of the ribosome could be identified, this result alone would not be sufficient to
determine the physiological significance of Hab1-dependent degradation and
further laborious experiments would be necessary. Further, the large amount of
data resulting from such analyses would distract from the clear message and
complete story we present in this paper. We therefore would prefer to leave such
questions for a subsequent study.

As an alternative approach, we present data demonstrating the
importance of the Hab1-ribosome interaction by showing that the M58A
mutation in Hab1, which decreases interaction with the ribosome, abolishes
autophagic delivery of the ribosome by Hab1. Further, we have added a cargo
analysis of Hab1 knockout cells versus those expressing Hab1, Hab1(M58A) or
Hab1(43-144) variants to demonstrate the necessary domains for Hab1-mediated
autophagy of ribosome. These data are now provided in Fig EV7 of the main text
(Line 269) in the revised manuscript.

Since we did not analyze total cell lysates in Fig 5C, we could not
perform the same analysis as shown in Fig 1C. We reanalyzed the Hab1 cargo
(Fig 5F) for proteins detected in the IP/MS analysis of Hab1(43-144) (Fig EV6A
in the revised manuscript) and found no non-ribosomal proteins that were
consistently delivered to the vacuole in a Hab1-dependent manner (Fig EV6E in
the revised manuscript). We have mentioned this in the manuscript (Line 224).

4. The requirement of Atg24 for efficient degradation of full-length Hab1 (Fig.
5C) is attributed to the latter's interaction with ribosomes. However, other large
cargos (Fas1, proteasomal particles) also require Atg24 for autophagic
elimination. It is, therefore, unclear whether endogenous elimination of
full-length Hab1 is mediated by ribosomes, other Atg24-dependent cargo, or
another Atg24-dependent mechanism. Therefore, the precise localization of
full-length Hab1 on wildtype and Atg24-deficient phagophores and its
colocalization and physical interaction with Atg24 should be experimentally
addressed.

It has recently been reported that Atg24 is required for the non-selective
sequestration of cargoes larger than 25 nm, as well as the complete opening of
the mouth of the autophagosome. (Kotani *et al.*, 2023; PMID: 37726301). As
noted by the referee, Hab1 binds ribosomes and is not degraded in the absence of
Atg24. For other large substrates such as Fas1 or proteasomal subunit proteins,
we were unable to detect any interaction with the C-terminal region of Hab1 (Fig.
EV6A in the revised manuscript), and a point mutation in M58 of Hab1 was
found to abolish both its interaction with the ribosome and the dependency of
degradation on Atg24 (Fig EV7D in the revised manuscript). These results
strongly suggest that the binding of Hab1 to ribosomes in particular, which are
large macromolecules that are not able to be isolated within autophagosomes in
the absence of Atg24, is the reason for the decreased delivery of Hab1 in
Atg24-deficient cells. Further, even if binding between Hab1 and Atg24 were
demonstrated, this result would not provide any insights into the mechanism of
Hab1 delivery.

Regarding the localization of Hab1 in *atg24*Δ cells, we have collected
data showing that Hab1 delivery to vacuoles was decreased by the disruption of
ATG24 and that Hab1 is sometimes localized on tubular-shaped structures (see
the attached panel C). This morphology is similar to that of isolation membranes
observed in *atg24*Δ cells (Kotani *et al.*, 2023), suggesting that Hab1 localizes to
the outer side of the isolation membrane.

At the time of submission of the previous manuscript, the paper on

Atg24 (Kotani *et al.*, 2023) had not been published. We have amended the
manuscript to include a description of these new findings regarding Atg24
function in the revised manuscript, which we feel should provide further
important context regarding the relationship between Atg24 and Hab1.

Panel for referee#3

**Attached figure**

(A) Rates of mitochondrial delivery to the vacuole were examined by GFP cleavage of Om45-GFP. Hab1(1-42) was tethered to mitochondria using the ALFA-tag system. -N, nitrogen starvation; mSc, mScarlet.

(B) Immunoblotting of subcellular fractionated lysates of cells expressing Hab1-GFP under control of endogenous or *TEF1* promoter. Lysates were obtained from cells treated with rapamycin for 0 or 4 h. Total (T), supernatant (S), and pellet (P) fractions. Rpl23 is a ribosomal protein.

(C) Confocal fluorescence imaging of WT or *atg24* Δ cells expressing Hab1-GFP and Vph1-2xmCherry. Scale bar, 5 μ m

Dear Dr. Ohsumi,

Thank you for submitting a revised version of your manuscript. Your study has now been seen by all original referees, who find that their previous concerns have been addressed and now recommend publication of the manuscript. There remain only a few mainly editorial points that have to be addressed before I can extend formal acceptance of the manuscript:

1. Please remove the AC/CRedit: section needs to be removed.
2. Table S5 is not referenced yet in the manuscript text and would need a callout.
3. Please upload the main and EV figures individually as high-res figure files.
4. Please reorganize the data files to one file/folder per figure and ZIPing for each main figure.
5. Synopsis:
Papers published in The EMBO Journal are accompanied online by a 'Synopsis' to enhance discoverability of the manuscript. It consists of A) a short (1-2 sentences) summary of the findings and their significance, B) 3-4 bullet points highlighting key results and C) a synopsis image that is 550x300-600 pixels large (width x height, jpeg or png format). You can either show a model or key data in the synopsis image. Please note that the image size is rather small, and that text needs to be readable at the final size. Please send us this information together with the revised manuscript.
6. Please add the accession ID for the JPOST repository is not provided in the data availability statement.
7. Figure Legends (main + EV): Please indicate the statistical test used for data analysis in the legends of figures 5d, f; EV 6b.
8. Please note that the box plots need to be defined in terms of minima, maxima, centre, bounds of box and whiskers, and percentile in the legends of figures 5d, f; EV 7e.
9. Please note that information related to n is missing in the legends of figures 5a-b, d, f; 6e-f; EV 7e; EV 9a-b.
10. Please note that the error bars are not defined in the legends of figures 4b-c; 6e-f; Ev 9a-b.
11. Please note that the measure of center for the error bars needs to be defined in the legends of figures 2a, d; 3b; 4a; 5a-b."
12. Please note that the arrowheads are not defined in the legend of figure EV 1c. This needs to be rectified."
13. There are 9 EV figures - some of them should be compiled in Appendix PDF with ToC with page numbers and nomenclature Appendix Figure Sx, and appropriate callouts
14. Tables S1-S5 should be renamed to Table EV1-EV5 with the corresponding callouts, and uploaded as Expanded View Content. Legends should be removed from ms file and included in the respected Excel files
15. Main and EV figure legends should be moved below the References

With best regards,

Cornelius Schneider

Cornelius Schneider, PhD
Editor
The EMBO Journal
c.schneider@embojournal.org

- a point-by-point response to the referees' comments, with a detailed description of the changes made (as a word file).
 - a word file of the manuscript text.
 - individual production quality figure files (one file per figure)
 - a complete author checklist, which you can download from our author guidelines (<https://www.embopress.org/page/journal/14602075/authorguide>).
 - Expanded View files (replacing Supplementary Information)
- Please see out instructions to authors
<https://www.embopress.org/page/journal/14602075/authorguide#expandedview>

We realize that it is difficult to revise to a specific deadline. In the interest of protecting the conceptual advance provided by the work, we recommend a revision within 3 months (12th May 2024). Please discuss the revision progress ahead of this time with the editor if you require more time to complete the revisions. Use the link below to submit your revision:

Referee #1:

I thank the authors for answering my questions and comments in a very satisfactory manner in their revised version, and congratulate them with a very nice and interesting paper.

Referee #2:

The authors have addressed all my concerns and I have no more comments. Congratulations to a very insightful study.

Referee #3:

The authors successfully addressed the concerns raised by the other reviewer, and in its present form, the manuscript meets EMBO J. merit.

All editorial and formatting issues were resolved by the authors.

Dear Prof. Ohsumi,

I am pleased to inform you that your manuscript has been accepted for publication in the EMBO Journal.

Yours sincerely,

Cornelius Schneider, PhD
Editor
The EMBO Journal
c.schneider@embojournal.org
